# Spatial alanine metabolism determines local growth dynamics of *Escherichia coli* colonies

**Francisco Díaz-Pascual[1], Martin Lempp[1], Kazuki Nosho[1], Hannah Jeckel[1,2,3], Jeanyoung K Jo[4], Konstantin Neuhaus[1,2,3], Raimo Hartmann[1], Eric Jelli[1,2], Mads Frederik Hansen[1], Alexa Price-Whelan[4], Lars EP Dietrich[4], Hannes Link[1,5], Knut Drescher[1,2,3]***

[1]Max Planck Institute for Terrestrial Microbiology, Marburg, Germany; [2]Department of Physics, Philipps-Universität Marburg, Marburg, Germany; [3]Biozentrum, University of Basel, Basel, Switzerland; [4]Department of Biological Sciences, Columbia University, New York, United States; [5]Interfaculty Institute for Microbiology and Infection Medicine, Eberhard Karls Universität Tübingen, Tübingen, Germany

**Abstract:** Bacteria commonly live in spatially structured biofilm assemblages, which are encased by an extracellular matrix. Metabolic activity of the cells inside biofilms causes gradients in local environmental conditions, which leads to the emergence of physiologically differentiated subpopulations. Information about the properties and spatial arrangement of such metabolic subpopulations, as well as their interaction strength and interaction length scales are lacking, even for model systems like *Escherichia coli* colony biofilms grown on agar-solidified media. Here, we use an unbiased approach, based on temporal and spatial transcriptome and metabolome data acquired during *E. coli* colony biofilm growth, to study the spatial organization of metabolism. We discovered that alanine displays a unique pattern among amino acids and that alanine metabolism is spatially and temporally heterogeneous. At the anoxic base of the colony, where carbon and nitrogen sources are abundant, cells secrete alanine *via* the transporter AlaE. In contrast, cells utilize alanine as a carbon and nitrogen source in the oxic nutrient-deprived region at the colony mid-height, *via* the enzymes DadA and DadX. This spatially structured alanine cross-feeding influences cellular viability and growth in the cross-feeding-dependent region, which shapes the overall colony morphology. More generally, our results on this precisely controllable biofilm model system demonstrate a remarkable spatiotemporal complexity of metabolism in biofilms. A better characterization of the spatiotemporal metabolic heterogeneities and dependencies is essential for understanding the physiology, architecture, and function of biofilms.

**\*For correspondence:**
knut.drescher@unibas.ch

**Competing interest:** The authors declare that no competing interests exist.

## Editor's evaluation

In this manuscript the authors characterize the temporal and spatial distribution of cells with different metabolic states within colony of biofilms of the model bacterium *Escherichia coli* using a combination of transcriptomics, metabolomics, and quantitative measurements of growth. They show that within the biofilm cells performing different metabolic functions are distributed in different regions of the colonies, and propose a model where nutrient cross-feeding through the amino acid alanine explains the phenotypic heterogeneity observed within the biofilm. Interestingly, they show that mutants impaired in cross-feeding have a fitness disadvantage. The finding reported and the innovative technical approaches used, can potentially be applied to identify novel metabolic interactions between communities of bacteria and understand how bacterial subpopulations interact spatially and metabolically within biofilms.

## Introduction

After bacterial cell division on surfaces, daughter cells often remain in close proximity to their mother cells. This process can yield closely packed populations with spatial structure, which are often held together by an extracellular matrix. Such spatially structured assemblages, called biofilms (*Flemming et al., 2016*), are estimated to be the most abundant form of microbial life on Earth (*Flemming and Wuertz, 2019*). The metabolic activity of cells inside these dense populations leads to spatial gradients of oxygen, carbon, and nitrogen sources, as well as many other nutrients and waste products (*Ackermann, 2015*; *Evans et al., 2020*; *Pacheco et al., 2019*; *Stewart and Franklin, 2008*). Cells in different locations within biofilms therefore inhabit distinct microenvironments. The physiological responses to these microenvironmental conditions result in spatially segregated subpopulations of cells with different metabolism (*Barroso-Batista et al., 2020*; *D'Souza et al., 2018*; *Stewart and Franklin, 2008*). Bacterial growth into densely packed spatially structured communities, and metabolic activity of the constituent cells, therefore naturally lead to physiological differentiation (*Evans et al., 2020*; *Røder et al., 2020*; *Serra et al., 2013a*; *Stewart and Franklin, 2008*).

Metabolic and phenotypic heterogeneities are frequently observed in multi-species communities (*Garg et al., 2016*; *Henson et al., 2019*; *Kim et al., 2020*; *Pacheco et al., 2021*; *Serra and Hengge, 2021*) and in single-species biofilm populations (*Cole et al., 2015*; *Dal Co et al., 2019*; *Lin et al., 2018*; *Moree et al., 2012*; *Rani et al., 2007*; *Serra et al., 2013b*; *Teal et al., 2006*). Identifying the origins of these heterogeneous subpopulations, and how they interact with each other, is important for understanding the development and function of biofilms (*Cole et al., 2015*; *Lin et al., 2018*; *Liu et al., 2015*; *Prindle et al., 2015*). Multi-species biofilms predominate in the environment, yet they are highly complex and feature many concurrent intra-species and inter-species interaction processes, which may be organized in space and time (*Brislawn et al., 2019*; *Garg et al., 2016*; *Kim et al., 2020*). Due to this complexity of multi-species biofilms, it is often difficult to disentangle whether spatiotemporal physiological differentiation or a particular interaction between subpopulations is caused by the relative position of the species, or the position of each species in the context of the entire community, or the mutual response of different species to each other. In contrast, single-species biofilms offer precisely controllable model systems with reduced complexity for understanding basic mechanisms of metabolic differentiation and the interaction of subpopulations. Investigations of phenotypic heterogeneity in single-species assemblages have already revealed fundamental insights into metabolic interactions of subpopulations with consequences for the overall fitness and growth dynamics of the assemblages (*Arjes et al., 2021*; *Cole et al., 2015*; *Evans et al., 2020*; *Lin et al., 2018*; *Liu et al., 2017*; *Liu et al., 2015*; *Wolfsberg et al., 2018*). However, even for single-species biofilms, the extent of metabolic heterogeneity and metabolic dependencies of subpopulations are unclear.

To obtain an unbiased insight into the spatial organization of metabolism inside biofilms, we measured metabolome and transcriptome dynamics during the development of *E. coli* colony biofilms on a defined minimal medium that was solidified with agar. Our model system enabled highly reproducible colony growth and precise control of environmental conditions, which allowed us to detect phenotypic signatures of subpopulations. The temporally and spatially resolved data revealed that alanine metabolism displays a unique pattern during colony growth. We determined that secretion of alanine occurs in a part of the anoxic region of the colony, where the carbon and nitrogen sources are abundant. The secreted alanine is then consumed in a part of the oxic region of the colony, where glucose and ammonium from the minimal medium are lacking. This spatially organized alanine cross-feeding interaction occurs over a distance of tens of microns, and has important consequences for the viability and growth of the localized cross-feeding-dependent subpopulation, and for the global colony morphology.

## Results

### Colony growth transitions and global metabolic changes

To investigate the spatiotemporal organization of metabolism inside *E. coli* colonies, we first characterized the basic colony growth dynamics and morphology on solid M9 minimal medium, which contained glucose and ammonium as the sole carbon and nitrogen sources, respectively (*Figure 1A*). For all measurements performed with colonies, including microscopy-based measurements, the colonies were grown on top of filter membranes that were placed on M9 agar (*Figure 1—figure*

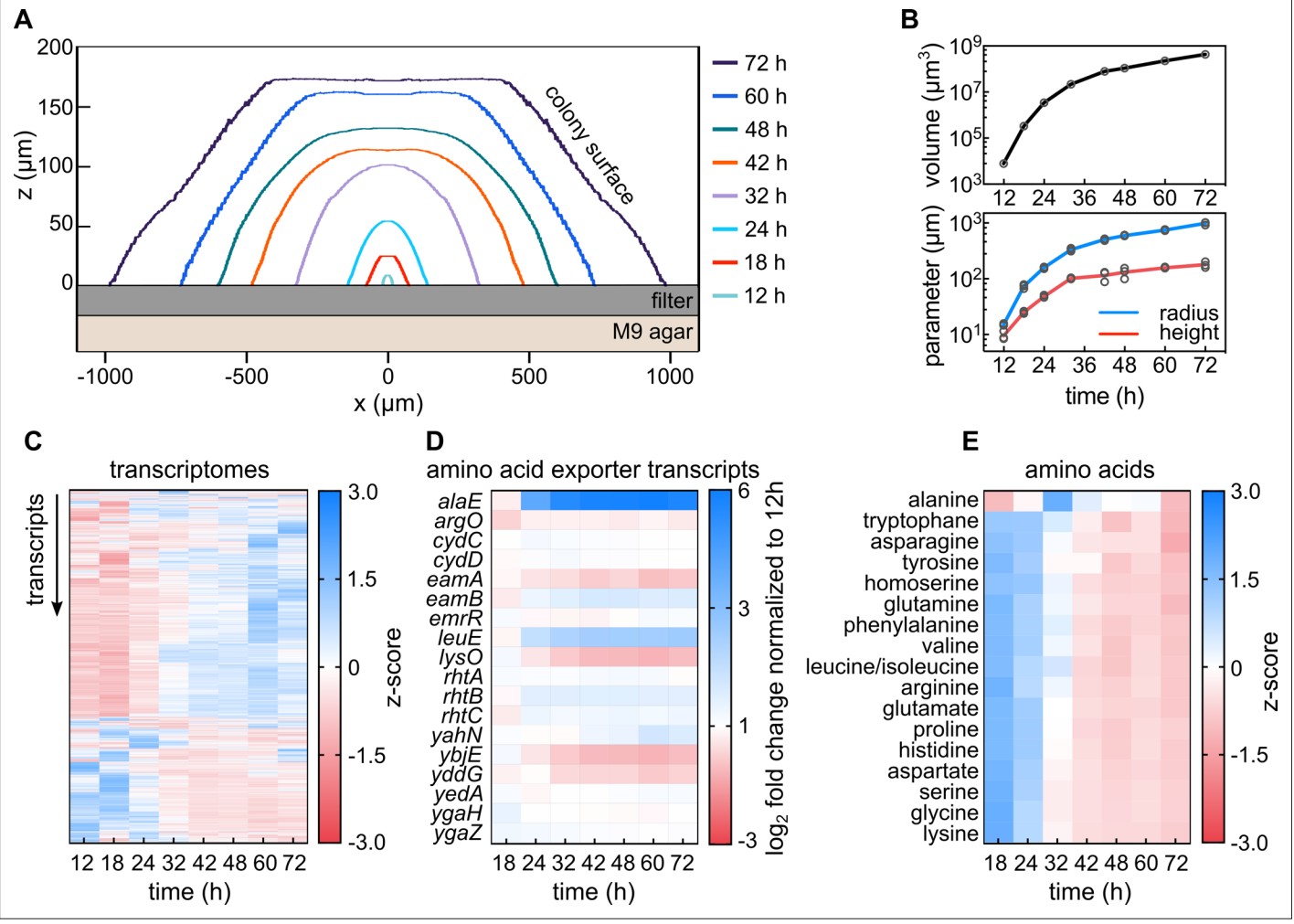

**Figure 1.** Transcriptomes and metabolomes during *E. coli* colony biofilm growth reveal metabolic transition and a unique role of alanine. (**A**) Cross-sectional profiles of representative *E. coli* colonies grown on membrane filters placed on solid M9 agar at different time points. (**B**) Volume (top panel), radius and height (bottom panel) of colonies as a function of time indicate a growth rate transition around 24–32 hr. All replicates are shown as individual data points, with a line connecting the mean values, n = 3 biological replicates. (**C**) Dynamics of expression profiles of 4231 genes during colony growth reveal physiological transition between 24 and 32 hr, n = 3 biological replicates. (**D**) Mean expression fold-changes of known amino acid exporters during colony growth, calculated in comparison to 12 hr colonies, n = 3 biological replicates. (**E**) Comparison of amino acid profiles in whole colonies, measured with mass spectrometry during colony growth; n = 3 biological replicates. Levels of amino acids are shown in *Figure 1—figure supplement 5*.

The online version of this article includes the following source data and figure supplement(s) for figure 1:

**Source data 1.** Source data for *Figure 1*.

**Figure supplement 1.** In contrast to wild-type colonies, flagella-deficient (Δ*fliC*) colonies are highly symmetric in shape.

**Figure supplement 2.** Principal component analysis of transcriptomic and metabolomic data.

**Figure supplement 3.** *E. coli* biofilm colonies express extracellular matrix genes.

**Figure supplement 4.** Spatial and temporal expression of genes involved in mixed acid fermentation and TCA cycle in colonies on M9 agar.

**Figure supplement 5.** Amino acid levels during colony growth.

---

*supplement 1A*). These membranes enabled rapid transfer of the colonies into extraction buffers, for downstream transcriptome and metabolome analyses. When we grew *E. coli* wild-type colonies in these conditions, we observed motility on the filter membranes on M9 agar, which led to heterogeneous colony development and an asymmetric colony morphology (*Figure 1—figure supplement 1B*). However, to study the spatiotemporal metabolism in *E. coli* colonies as precisely as possible, and to resolve even small phenotypic differences between strains and conditions, we required a colony

growth behavior that is as reproducible as possible. Therefore, a strain lacking flagella due to the deletion of the flagellin (ΔfliC) was used as the parental strain for all following experiments, which resulted in highly reproducible and axially symmetric colony morphologies (*Figure 1—figure supplement 1C*). Colonies generally displayed two growth phases: colonies showed exponential growth in volume, height, and diameter for up to ~24 hr, followed by linear growth (*Figure 1A and B*). This transition in colony growth dynamics has previously been observed, and was hypothesized to be caused by a change in metabolism due to altered nutrient penetration and consumption for colonies above a certain size (*Pirt, 1967*; *Warren et al., 2019*).

To further characterize the phenotypic changes that occur during colony growth, we performed time-resolved whole-colony transcriptome measurements (*Figure 1C* and *Figure 1—figure supplement 2A*). The transcriptomes revealed a major change after 24 hr of growth (*Figure 1C*), which reflects the change in growth dynamics that was apparent in the morphological parameters (*Figure 1B*). After 72 hr of growth, 966 out of 4231 detected genes were differentially expressed in comparison to the 12 hr time point (log₂-fold-changes > 1 or <-1, and FDR-adjusted p-value < 0.05). The whole-colony transcriptomes also showed that biofilm matrix biosynthesis genes are expressed continuously during colony growth (*Figure 1—figure supplement 3*). Furthermore, the transcriptomes confirmed that well-known pathways such as mixed acid fermentation and tricarboxylic acid (TCA) cycle were differentially regulated after 24 hr of growth (*Figure 1—figure supplement 4A,C*). These results are consistent with the hypothesis that above a certain colony size, which is reached between 24 and 32 hr in our conditions, the consumption of oxygen by cells in the outer region of the colony causes a large anoxic region inside the colony.

Although it is commonly assumed that *E. coli* subpopulations cross-feed acetate (*Cole et al., 2015*; *Dal Co et al., 2019*; *San Roman and Wagner, 2020*; *Wolfsberg et al., 2018*), the temporal transcriptomes did not reveal strong regulation of acetate metabolism transcripts during the major transition in colony metabolism described above (*Figure 1—figure supplement 4A*). However, transcripts of the lactate, formate, and succinate biosynthesis pathways displayed differential regulation during colony growth (*Figure 1—figure supplement 4A*). Apart from these metabolites, we noticed peculiar patterns in the expression of amino acid pathways. The gene expression levels for some amino acid transporters remained unchanged during colony growth, while a few amino acid transporters were >2 fold up- or down-regulated after 24 hr or 32 hr of growth (*Figure 1D*). Interestingly, the gene coding for the alanine exporter AlaE (*Hori et al., 2011b*) showed strong expression changes (~50 fold increase) in 72-hr-colonies relative to 12-hr-colonies (*Figure 1D*).

To further characterize the observed metabolic transition between 24 and 32 hr of colony growth, we measured the amino acid profiles of the colonies using mass spectrometry and normalized them by the colony biomass. To obtain sufficient biomass for these measurements, the colonies needed to be grown for at least 18 hr. All amino acid abundances decreased during colony growth – except for alanine, which remained relatively constant with a peak abundance at 32 hr (*Figure 1E*, *Figure 1—figure supplement 2B*, and *Figure 1—figure supplement 5*). Thus, both transcriptome and metabolome data suggest a unique change of alanine metabolism during *E. coli* colony growth, which led us to investigate the role of alanine metabolism in colony morphogenesis.

## Spatial regulation of alanine transport and degradation

Biofilms are expected to be metabolically heterogeneous so that we hypothesized that alanine metabolism is spatially organized inside biofilms. To test this hypothesis, we developed a method to measure transcriptomes with spatial resolution in the colonies. The method is based on the oxygen-dependence of chromophore maturation of the fluorescent protein mRuby2 (*Lam et al., 2012*; *Tsien, 1998*). Using a strain that constitutively expresses mRuby2 from a chromosomal locus, so that all cells in the colony produce mRuby2, we observed that only the air-facing region of the colony was fluorescent in colonies that had grown for 72 hr. Quantification of the mRuby2 fluorescence showed a similar, but slightly steeper decrease of fluorescence in the horizontal *xy*-direction into the colony, compared with the vertical *z*-direction (*Figure 2A*). By using an oxygen microsensor to directly measure oxygen levels inside the colony in the *z*-direction (*Jo et al., 2017*), we determined that the mRuby2 fluorescence was a reliable indicator of oxygen levels (*Figure 2A*). During colony growth, the fraction of fluorescent cells in the colony decreased (inset in *Figure 2B*), and the majority of the colony became non-fluorescent (i.e. anoxic) around 24 hr, which coincides with the time at which the whole-colony transcriptome

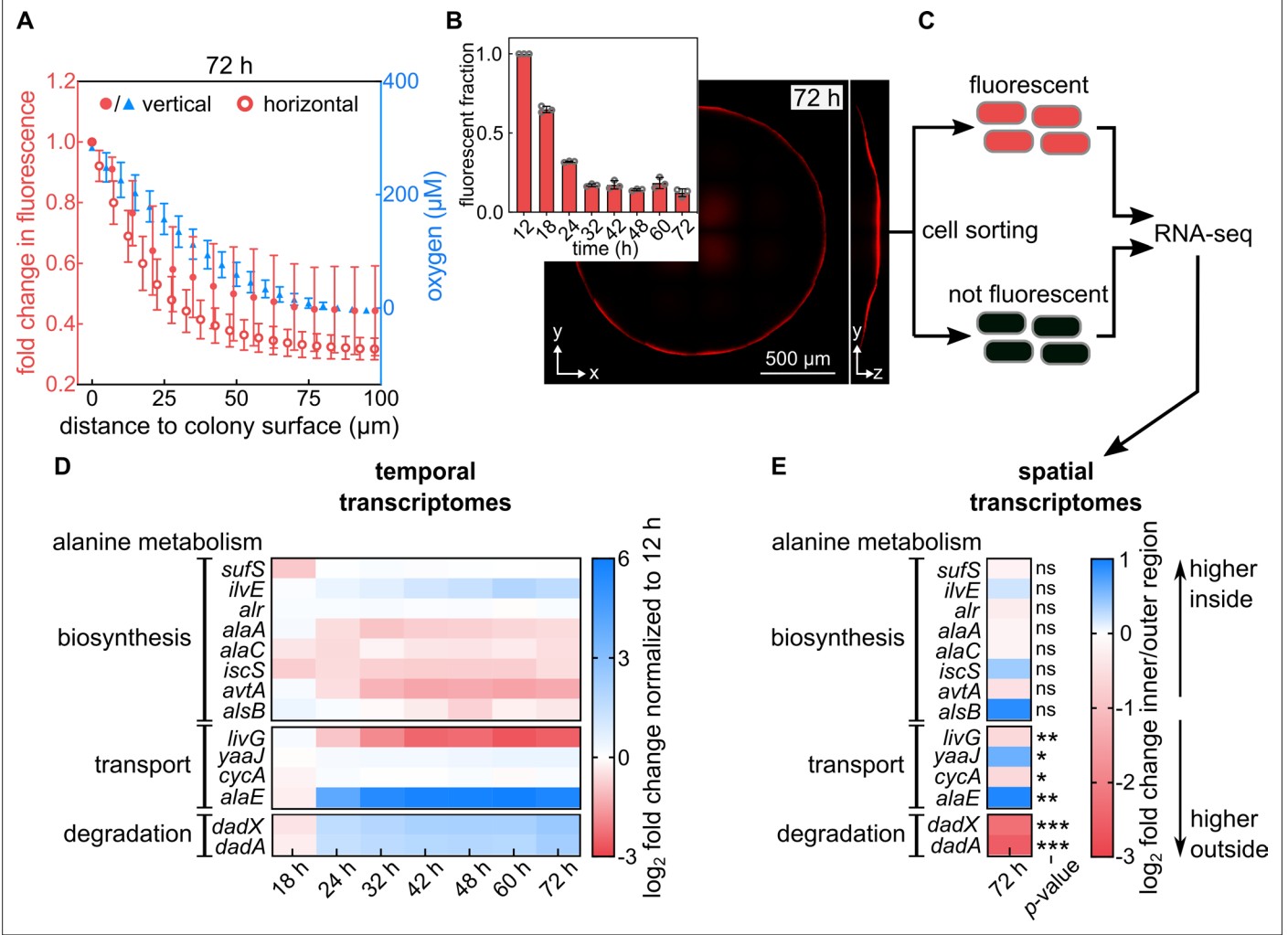

**Figure 2.** Alanine transport and degradation are spatially regulated within colony biofilms. (**A**) Measurement of oxygen penetration into colonies grown for 72 hr on filter membranes on M9 agar using two different methods (for strain KDE722). Left axis (red): Intensity of mRuby2 fluorescence within a vertical cylinder of radius 3.3 µm in the center of the colony (filled circles), and intensity of mRuby2 fluorescence within a horizontal plane at the base of the colony (open circles, mean ± s.d., n = 3 biological replicates). Right axis (blue): Direct measurement of oxygen levels acquired by vertical scanning of an oxygen microsensor at the center of the colony (blue triangles, mean ± s.d., n = 10 replicates). (**B**) Confocal image of a representative 72 hr colony of a strain that constitutively expresses mRuby2 (KDE722). Insert: Fraction of fluorescent biovolume inside colonies grown for different times. Data are mean ± s.d., n = 3 biological replicates. (**C**) Scheme of the sorting procedure: Cells from a 72 hr colony are separated, fixed using formaldehyde, and sorted according to their mRuby2 fluorescence, followed by RNA-seq, as described in the methods section. (**D**) Heatmaps showing the mean fold-change (n = 3 biological replicates) of expression levels of genes involved in alanine metabolism, from whole-colony measurements. Fold-changes are computed relative to the 12 hr timepoint. (**E**) Spatial transcriptome results, quantified as fold-changes between the inner region (no mRuby2 fluorescence) and outer region (high mRuby2 fluorescence) regions of colonies grown for 72 hr. Blue color in the heatmap indicates genes with higher transcript levels in the inner region of the colony, red color indicates higher transcripts in the outer region of the colony. Data are means, n = 4 biological replicates. Non-significant differences between spatial regions are labeled 'ns'. The p-values correspond to false discovery rate (FDR)-adjusted p-values: * p < 0.05, ** p < 0.01, *** p < 0.001.

The online version of this article includes the following source data and figure supplement(s) for figure 2:

**Source data 1.** Source data for *Figure 2*.

**Figure supplement 1.** Fluorescent gradients of mRuby2 in growing colonies are not due to imaging artefacts.

**Figure supplement 2.** Formaldehyde fixation does not affect mRuby2 fluorescence or the colony transcriptome.

**Figure supplement 3.** Application of flow cytometry with fluorescence-activated cell sorting to separate cells from the inner and outer regions of colonies.

**Figure supplement 4.** Key pathways in alanine metabolism.

shifted towards anaerobic metabolism (*Figure 1C*). This decrease in the mRuby2-fluorescent population during colony growth ultimately led to a thin layer of fluorescent cells in the air-facing part of the colony (*Figure 2B*).

To eliminate the possibility that the observed mRuby2 fluorescence profile was due to imaging artefacts, such as insufficient laser penetration into the colony, we disrupted 72-hr-colonies and imaged the resulting, well-separated single cells. The time between colony disruption and imaging for this control experiment was less than 2 min, which is substantially shorter than the mRuby2 fluorophore maturation time (~150 min *Lam et al., 2012*). In these images, only some cells displayed fluorescence (*Figure 2—figure supplement 1A*), indicating that the fluorescence gradient we observed in *Figure 2B* is not an imaging artefact. In an additional control experiment, we sought to test if the fluorescence gradient was caused by the oxygen gradient. If the oxygen gradient in the colony is created by oxygen consumption by cells in the air-facing region (*Klementiev et al., 2020*; *Stewart and Franklin, 2008*), the oxygen gradient (and therefore the mRuby2 fluorescence gradient) should disappear when metabolic processes that consume oxygen are prevented. To test this, we starved the colonies by transferring the filter membrane carrying the colonies to an M9 agar plate lacking glucose, which strongly decreases the colony capacity to consume oxygen. We observed that in this case, mRuby2 proteins that are located in the formerly dark anoxic region of the colony became fluorescent (*Figure 2—figure supplement 1B,C,D*). The oxygen in the fresh agar plate is not able to cause the entire colony to become fluorescent without the reduced oxygen consumption in the colony caused by the lack of glucose. The finding that the entire colony becomes fluorescent after the transfer is consistent with the interpretation that in the absence of glucose, cells consume less oxygen so that molecular oxygen can penetrate into the colony to enable chromophore maturation of mRuby2 in the formerly anoxic region (*Figure 2—figure supplement 1B,C,D*). Together, these control experiments and the direct measurements of oxygen levels inside the colonies show that the difference in fluorescence levels in our system reflect the spatial position of the cells in the colony.

To obtain spatial transcriptomes, we then separated colonies grown for 72 hr into individual cells, immediately fixed them with formaldehyde (which prevents mRuby2 fluorophore maturation in the presence of oxygen without altering the transcriptomes; *Figure 2—figure supplement 2*), and subjected them to fluorescence-activated cell sorting (FACS) to separate the oxic (fluorescent) and anoxic (not fluorescent) populations (*Figure 2C* and *Figure 2—figure supplement 3*). The resulting two cell populations were then analyzed using RNA-seq. The spatial transcriptome comparison showed that, as expected, genes involved in the TCA cycle and mixed acid fermentation were differentially expressed between the inner and outer regions of the colony (*Figure 1—figure supplement 4B,D*). This result demonstrates that the method successfully separated the fermenting population inside the colonies from the respiring population in the outer layer, which serves as a qualitative verification of the experimental methodology.

In both the spatially and temporally resolved transcriptomes, we observed changes in alanine transport and degradation, but not in alanine biosynthesis (*Figure 2D and E*; a schematic diagram of alanine metabolism pathways is shown in *Figure 2—figure supplement 4*). In particular, the spatial transcriptomes showed that the expression of the alanine exporter gene *alaE* was significantly upregulated in the inner (non-fluorescent) region of the colony, compared with the outer (fluorescent) region (*Figure 2E*). Alanine conversion into pyruvate and ammonium was also spatially regulated. Two pathways for converting alanine to pyruvate are known: The reversible conversion by enzymes involved in the alanine biosynthetic pathways, and the irreversible conversion mediated by the *dadAX* operon (*Figure 2—figure supplement 4B*). The latter encodes a racemase (*dadX*) and a dehydrogenase (*dadA*) (*Lobocka et al., 1994*). In colonies grown for 72 hr, the *dadAX* operon was down-regulated in the inner region, compared with the outer region (*Figure 2E*). Interestingly, in the whole-colony temporal transcriptomes, the *dadAX* operon and the gene *alaE* were the only strongly upregulated alanine-related genes (sixfold for *dadA*, fivefold for *dadX*, and 50-fold for *alaE*), when comparing 72-hr-colonies to 12-hr-colonies (*Figure 2D*). These results indicate that colonies globally upregulate alanine export and degradation during development and that the anoxic region of the colony likely exports alanine, while the oxic region likely converts alanine into pyruvate and ammonium.

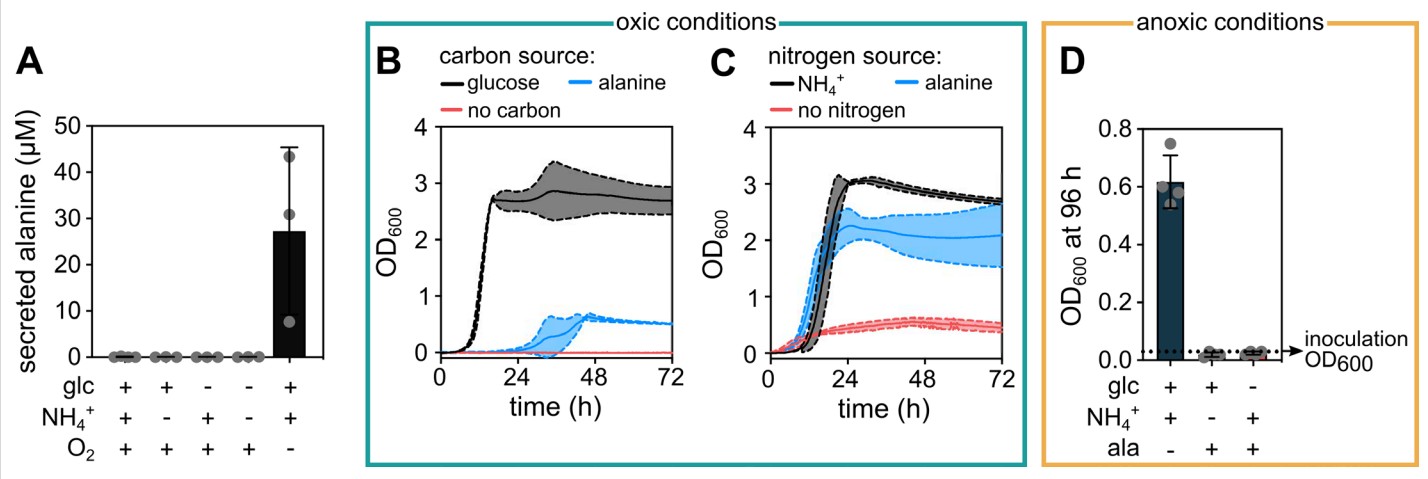

**Figure 3.** Alanine can be secreted in anoxic conditions and consumed in oxic conditions by *E. coli*. (**A**) Extracellular alanine concentration in the supernatant of liquid cultures grown in presence or absence (indicated by+ or -) of glucose (glc), ammonium (NH₄⁺), and molecular oxygen (O₂). Data are mean ± s.d., n = 3 biological replicates. (**B**) Growth curves using M9 minimal salts medium with ammonium as nitrogen source and different carbon sources: Either glucose (5 g/L, as in our standard M9 medium), alanine (10 mM), or no carbon source. (**C**) Growth curves using M9 minimal salts medium with glucose as a carbon source and different nitrogen sources: Either ammonium (22.6 mM, as in our standard M9 medium), alanine (5 mM) or no nitrogen source. For panels **B** and **C**, continuous middle lines correspond to the mean and the dotted lines to the standard deviation, n = 3 biological replicates. The growth curves shown for alanine in panels **B** and **C** correspond to the alanine concentrations that resulted in the highest final optical density at 600 nm (OD₆₀₀) in each condition. (**D**) *E. coli* cultures starting with OD₆₀₀ = 0.03 were incubated for 96 hr in different anoxic media. The medium contained combinations of glucose, NH₄⁺, and alanine as indicated. Alanine was provided either as the sole nitrogen source (middle bar) or sole carbon source (right bar) with a concentration of 5 mM and 10 mM, respectively. The final OD₆₀₀ after 96 hr of incubation is shown, corresponding to a time point when the culture without alanine reached stationary phase. Continuous measurements of OD₆₀₀ in anoxic conditions was not possible in our laboratory due to technical limitations. Data are mean ± s.d., n = 4 biological replicates.

The online version of this article includes the following source data for figure 3:

**Source data 1.** Source data for *Figure 3*.

## Alanine is exported in anoxic conditions and can be used as a carbon and nitrogen source in oxic conditions

The spatial transcriptomes suggest that alanine is primarily secreted in the anoxic region of the biofilm. To test this, we explored under which combination of carbon/nitrogen/oxygen availability *E. coli* secretes alanine in shaking liquid conditions. Mass spectrometry measurements from culture supernatants clearly showed that alanine is only secreted under anoxic conditions with glucose and ammonium (*Figure 3A*), which is an environment that corresponds to the anoxic base of the colony, where cells are in contact with the glucose- and ammonium-rich M9 agar. Oxic conditions with abundant glucose and ammonium did not result in significant alanine secretion. This finding suggests that alanine is secreted in the anoxic base of the colony, which is consistent with the spatial transcriptome results.

The spatial transcriptomes also suggest that alanine is primarily consumed in the oxic region of the colony. It is well-known that *E. coli* can use extracellular alanine as a carbon or nitrogen source in oxic conditions (*Franklin and Venables, 1976*; *Kornberg, 1965*), which we confirmed for our strain and growth conditions by replacing either glucose or ammonium with alanine in the liquid shaking M9 medium (*Figure 3B and C*). We found that exogenous alanine can be utilized as a poor carbon source, but as a good nitrogen source in oxic conditions. Interestingly, we observed that under anoxic conditions, *E. coli* cannot utilize alanine as a carbon or nitrogen source (*Figure 3D*). Together, these results suggest that the alanine secreted in the glucose- and ammonium-rich anoxic region of the colony can be consumed in the oxic region of the colony as a cross-fed metabolite.

## Bacterial survival in the oxic region of the colony is influenced by alanine export and consumption

To determine how interference with alanine export and consumption affects colony growth, we created individual and combinatorial deletions of known alanine transport and degradation genes. None of these deletions affected the cellular growth rates in liquid culture (*Figure 4—figure supplement 1A*). These deletions also did not cause clear phenotypes in colony height or diameter after

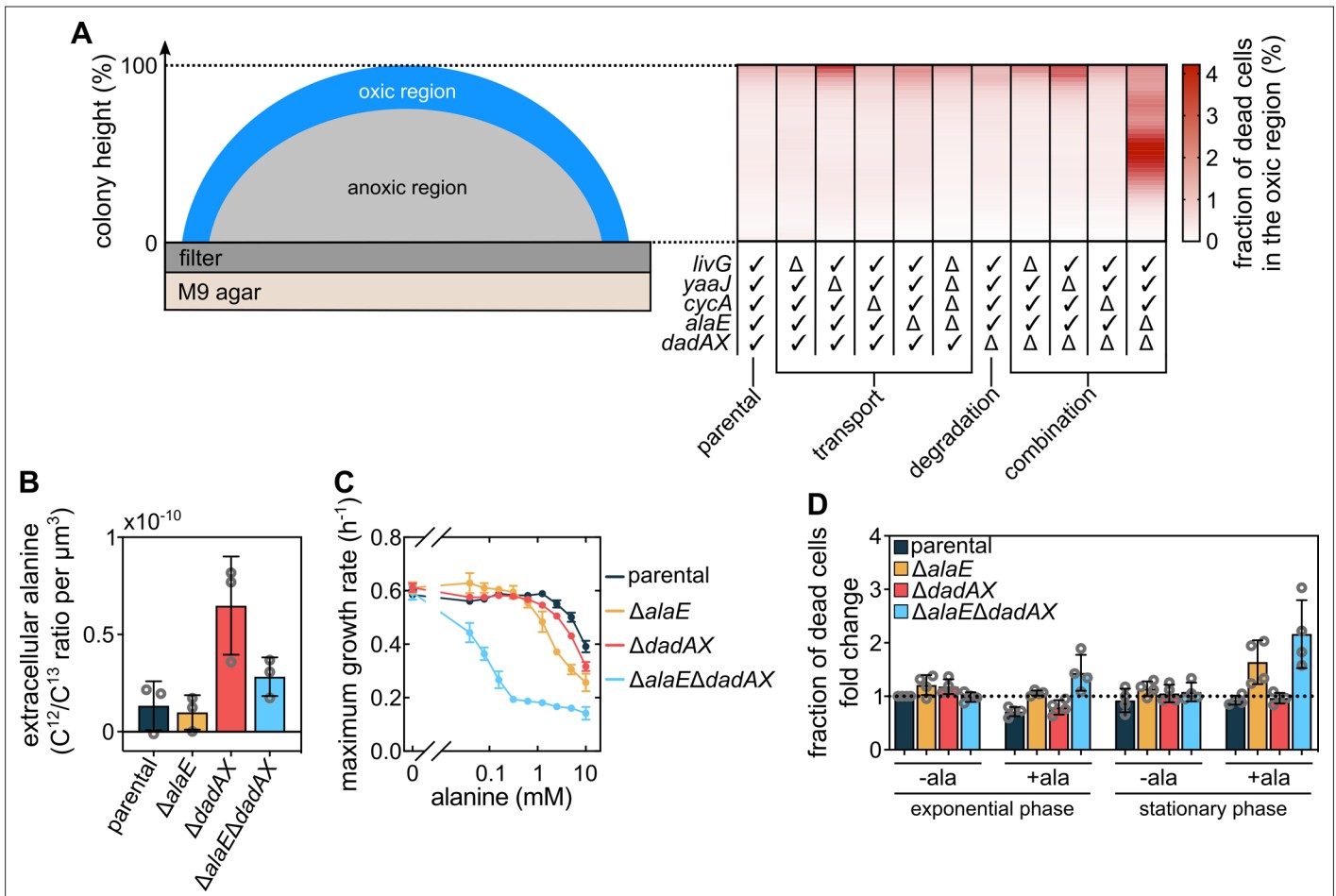

**Figure 4.** Alanine influences cell viability and growth. (**A**) Measurement of the fraction of dead cells as a function of height in colonies grown for 72 hr on M9 agar. These measurements were performed only for cells located within 30 μm from the outer colony surface, which is a conservative measurement of the oxic region (*Figure 2A*). The table designates the genotype of strains that were investigated: '✓' indicates that the gene is intact and 'Δ' that the gene was deleted. Data in the heatmaps shows means, n = 3 biological replicates. Errors are shown in *Figure 4—figure supplement 2*. (**B**) Extracellular alanine levels measured from colonies (mean ± s.d., n = 3 biological replicates). (**C**) Maximum liquid culture growth rate (in presence of glucose and ammonium) for different strains, as a function of the concentration of exogenously added alanine. Data are mean ± s.d., n = 3 biological replicates. (**D**) Fraction of dead cells, measured using SYTOX Green fluorescence normalized by $OD_{600}$, in cultures grown with glucose and ammonium in presence or absence of 5 mM alanine. Fold-changes are calculated relative to the parental strain in exponential phase without alanine. Measurements were performed when cultures reached half of the maximum $OD_{600}$ (for exponential phase measurements) or when they reached their maximum $OD_{600}$ (for stationary phase measurements). Bars indicate mean ± s.d., n = 4 biological replicates.

The online version of this article includes the following source data and figure supplement(s) for figure 4:

**Source data 1.** Source data for *Figure 4*.

**Figure supplement 1.** Growth rates and colony sizes for alanine metabolism mutants.

**Figure supplement 2.** Coefficient of variation of the fraction of dead cells in the oxic region of the colony.

**Figure supplement 3.** Effects of exogenously added alanine on maximum growth rate for alanine metabolism mutants are specific to alanine, and are not triggered by exogenously added serine.

**Figure supplement 4.** Cells capable of secreting and consuming alanine have a fitness advantage during colony growth.

72 hr of incubation on M9 agar (*Figure 4—figure supplement 1B,C*), indicating that alanine export and consumption do not have large effects on global colony size. However, the mutants displayed substantial differences when we measured the fraction of dead cells in the oxic region (*Figure 4A* and *Figure 4—figure supplement 2*), using a fluorescent nucleic acid stain that can only penetrate the disrupted membranes of dead cells (SYTOX Green). We limited our measurements to the air-facing oxic region of the colony, because in this region the SYTOX Green fluorescence can be reliably quantified without microscopy artefacts that may occur deeper inside the colony due to poor laser penetration. We observed that the colonies displayed only a low fraction of dead cells at the bottom edges of the colony, but cell death increased towards the top of the colony. Interestingly, the oxic region at around 50 % of the maximum colony height displayed increased cell death when cells carried the Δ*alaE*Δ*dadAX* deletion (*Figure 4A*), which is a strain that should have a strongly reduced capability for secreting and consuming alanine.

To determine if the increased cell death displayed by the Δ*alaE*Δ*dadAX* mutants could be caused by an impaired alanine cross-feeding, we first measured the extracellular alanine concentration in colonies of the relevant mutants using mass spectrometry (*Figure 4B*). As expected, mutants incapable of the major alanine degradation pathway (Δ*dadAX*) displayed substantially higher extracellular alanine than cells that are impaired for alanine secretion as well (Δ*alaE*Δ*dadAX*). Amino acid transporters can often function as both importers and exporters, yet the higher extracellular alanine levels of the Δ*dadAX* strain compared with the Δ*alaE*Δ*dadAX* strain (*Figure 4B*) indicate that AlaE acts as an alanine exporter inside colonies. Previous measurements have shown that AlaE also acts as an exporter in liquid cultures (*Hori et al., 2011b*). The high extracellular alanine levels of the Δ*dadAX* colonies are unlikely to be caused by permeable or lysed cells, as the Δ*dadAX* colonies do not display elevated levels of cell death (*Figure 4A*). The parental strain and the Δ*alaE* mutant displayed similarly low extracellular alanine levels, close to the detection limit of our mass spectrometry technique. Interestingly, and consistent with the results of Hori et al. in liquid cultures (*Hori et al., 2011a*; *Hori et al., 2011b*), our detection of extracellular alanine in colonies of the Δ*alaE* and Δ*alaE*Δ*dadAX* mutants indicates that another mechanism for alanine export might exist that is currently unknown. Together, our measurements of extracellular alanine levels in colonies of different mutant show that alanine is primarily secreted via AlaE.

For cells that lack both the major alanine exporter AlaE and the major alanine degradation pathway *via* DadA and DadX, we hypothesized that the presence of extracellular alanine might lead to an accumulation of intracellular alanine to toxic levels. It has previously been shown that excess levels of intracellular alanine can inhibit growth (*Katsube et al., 2019*), yet the molecular mechanism underlying this process is still unclear. Indeed, we observed that all strains show a decreased growth rate in liquid media containing high levels of alanine, yet strains carrying the Δ*alaE*Δ*dadAX* mutation were much more sensitive to exogenously added alanine than the parental strain (*Figure 4C*). This result was not due to unspecific effects of alanine (such as osmolarity changes), because no significant differences between the mutants and the parental strain were observed when serine was added exogenously instead of alanine (*Figure 4—figure supplement 3*). Therefore, extracellular alanine can modulate bacterial growth rates, particularly for strains that are deficient in alanine cross-feeding (Δ*alaE*Δ*dadAX*).

Since colonies can accumulate alanine in their extracellular space (*Figure 4B*) and the cellular growth rate can be reduced by extracellular alanine (*Figure 4C*), we hypothesized that the increased cell death in the oxic region of the Δ*alaE*Δ*dadAX* colonies (*Figure 4A*) is due to the accumulation of toxic extracellular alanine levels in this region, which arise from the impaired cross-feeding of this strain. To test this hypothesis, we measured cell viability for the parental strain and mutants in liquid cultures with and without exogenous alanine during mid-exponential phase and in stationary phase. We found that even though the parental strain displayed a reduced growth rate when exposed to extracellular alanine (*Figure 4C*), the parental strain did not display increased levels of cell death in such conditions (*Figure 4D*) – likely due to their ability to secrete and consume alanine, allowing them to control their intracellular levels of alanine. In contrast, we found that in the presence of high extracellular alanine concentrations, Δ*alaE*Δ*dadAX* mutants displayed higher cell death levels than the parental strain, which was particularly strong in stationary phase conditions (*Figure 4D*). The increased cell death of the Δ*alaE*Δ*dadAX* mutants was not accompanied by a reduction in optical density, indicating these cells did not lyse. If cells can still export alanine (Δ*dadAX*), cell viability in liquid cultures is significantly improved in the presence of extracellular alanine, compared with the

ΔalaEΔdadAX mutant (**Figure 4D**). If cells can still degrade alanine (ΔalaE), cell viability is only slightly improved under stationary phase conditions, compared with the ΔalaEΔdadAX mutant (**Figure 4D**). These cell viability measurements are consistent with the effect of extracellular alanine on the growth rate of ΔalaEΔdadAX and ΔalaE mutants (**Figure 4C**). We speculate that the exponentially growing cultures resembled the aerobic periphery at the base of the colony that is in contact with glucose and ammonium, whereas the stationary phase cultures resemble the oxic region above, which is nutrient depleted if no cross-feeding is present.

Together, these results support the hypothesis that in ΔalaEΔdadAX mutant colonies, extracellular alanine accumulates due to impaired cross-feeding, which causes the increased cell death compared with the parental strain. Alanine cross-feeding should therefore have an important effect for the oxic region above the nutrient-rich base of the colony.

## Strains impaired in alanine cross-feeding display a reduced fitness in colonies

If alanine cross-feeding has a significant impact on colony growth, we would expect that cross-feeding-impaired cells also have a fitness disadvantage compared to the parental strain during colony growth. To test if this is the case, we generated pairwise mixtures of different strains and inoculated these mixtures onto a membrane filter placed on M9 agar. It is important to note that this inoculation procedure using a liquid drop (leading to colonies that were inoculated by hundreds of cells) creates different colony morphologies compared to colonies grown from a single bacterium (which was the growth condition for all other experiments with colonies in this article). After 72 hr of incubation of the mixed-strain colonies, we compared the frequency of each strain at the growing front of the colony to the inoculation frequency (**Figure 4—figure supplement 4**). In a control experiment, we observed that for strains that were isogenic except for the fluorescent protein they express, the superfolder GFP (sfGFP) expressing strain slightly outcompeted the mRuby2 expressing strain – likely due to a difference in the energetic cost of the biosynthesis of each fluorescent protein. We also observed that the ΔalaEΔdadAX mutant was consistently strongly outcompeted by the parental strain (**Figure 4—figure supplement 4B**), despite having no liquid culture grow differences (**Figure 4—figure supplement 1**) and regardless of the fluorescent protein that they were carrying. These experiments indicate that cells capable of alanine cross-feeding have a fitness advantage during colony growth.

## Alanine cross-feeding influences colony morphology

If the alanine molecules that are secreted in the anaerobic base of the colony are consumed in the oxic higher regions of the colony, alanine could serve as a carbon and nitrogen source and support growth in this region. From measurements of the colony height and diameter for mutants impaired in cross-feeding, we know that alanine cross-feeding does not have a major influence on colony size (**Figure 4—figure supplement 1**). These measurements of colony height and diameter also show that alanine cross-feeding does not contribute to aerobic growth at the very top of the colony or at the outer edge of the base. We therefore investigated effects of alanine cross-feeding on cellular growth in the oxic region at mid-height, where the fraction of dead cells was highest for the ΔalaEΔdadAX mutants (**Figure 4A**). Measurements of the colony morphology showed that ΔalaEΔdadAX and ΔalaE mutants displayed a significantly decreased colony curvature in the relevant region, in comparison to the parental strain (**Figure 5A and B**). The detection of this phenotypic difference in colony morphology was enabled by the high reproducibility of our *E. coli* colony biofilm growth assay. The 'bulge' of the parental colonies suggests that this region grows due to alanine cross-feeding. This interpretation is consistent with recent simulations of colony growth without cross-feeding, which resulted in nearly conical colony shapes with triangular *xz*-cross sections (**Warren et al., 2019**) that lacked the 'bulge' morphology we observed for the cross-feeding capable strains.

To further experimentally investigate the effect of alanine cross-feeding on cellular growth in the mid-height oxic region, we measured the fluorescence ratio of an unstable version of the superfolder GFP (**Andersen et al., 1998**) and the long-lived mRuby2, both of which were expressed constitutively using a $P_{tac}$ promoter. The ratio of these fluorescent proteins serves as a measure of the cellular growth rate (**Figure 5—figure supplement 1**). These measurements showed that the parental strain grows faster in the mid-height oxic region than the cross-feeding impaired ΔalaEΔdadAX mutant

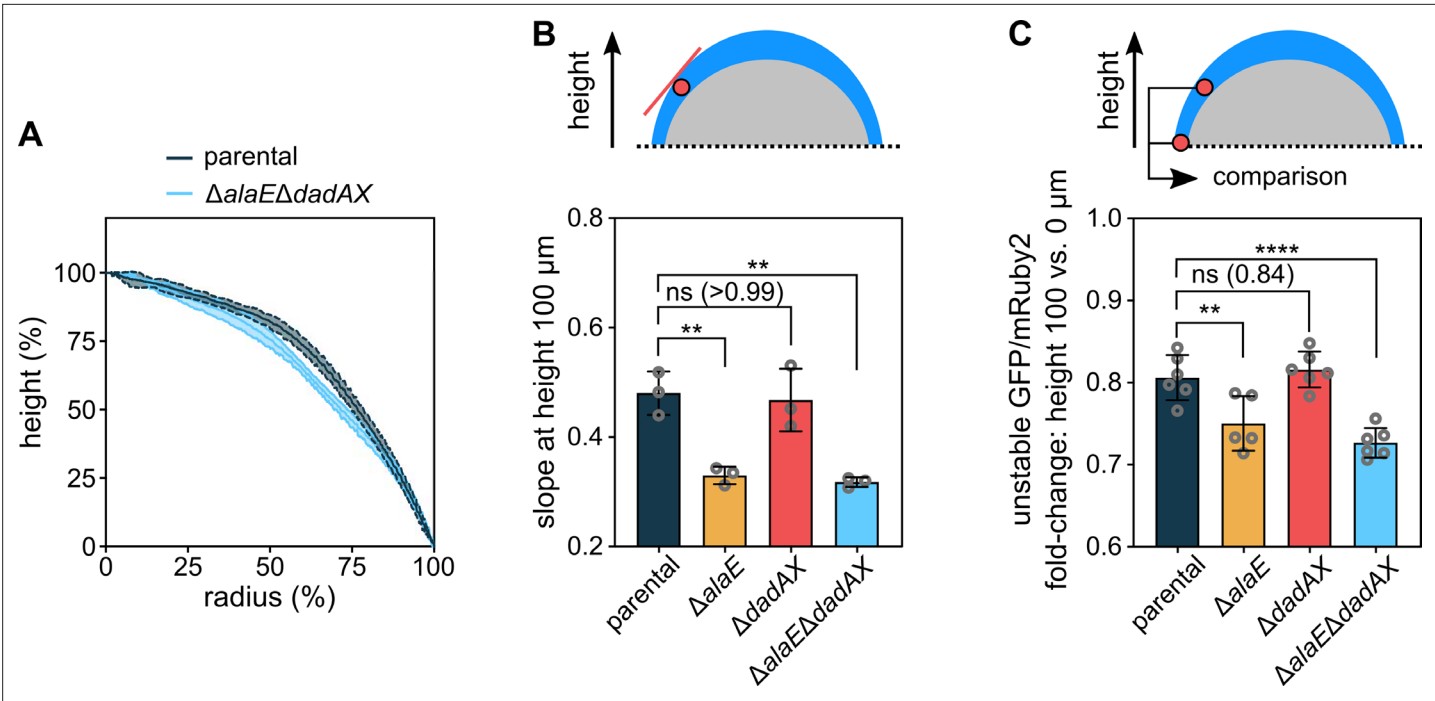

**Figure 5.** Colony morphology is influenced by alanine cross-feeding. (**A**) Colony height as a function of the colony radius, for colonies grown for 72 hr. Central lines correspond to the mean and the shaded area to the s.d., n = 3 biological replicates. (**B**) Slope of curves in panel *A* at height 100 µm. Data are mean ± s.d., n = 3 biological replicates. (**C**) Fluorescence ratio of constitutively expressed unstable GFP and stable mRuby2, which is a measure for the cellular growth rate, as shown in *Figure 5—figure supplement 1*. The unstable GFP was constructed by adding the ASV-tag to superfolder GFP. The fold-change of this fluorescent protein ratio was calculated for the oxic colony region between the heights 100 µm and 0 µm. The fluorescent protein ratio was only measured for cells located within 30 µm from the outer colony surface, as a conservative measure of the oxic region. Data are mean ± s.d., n = 5–6 biological replicates. Statistical significances were calculated using one-way ANOVA with Dunnett's correction. Non-significant differences are labeled 'ns' and the p-value is shown in brackets; **p < 0.01, ****p < 0.0001.

The online version of this article includes the following source data and figure supplement(s) for figure 5:

**Source data 1.** Source data for *Figure 5*.

**Figure supplement 1.** Ratio between unstable GFP and stable mRuby2 correlates with bacterial growth rate.

(*Figure 5C*), which further supports the hypothesis that this region of the colony relies on alanine as a carbon and nitrogen source.

## Discussion

We showed that alanine metabolism is spatiotemporally regulated inside *E. coli* colony biofilms. The high degree of control and reproducibility of our model system enabled us to determine that interference with the export and consumption of alanine causes phenotypes in cell viability and cell growth rate. These viability and growth rate phenotypes are localized in the region of the colony that depends on alanine as a nutrient source, but they affect the global morphology of the colony.

Based on our results, we propose the following model for the spatial organization of alanine metabolism in colonies that have grown for 72 hr (*Figure 6A*): Cells at the bottom periphery of the colony (red region in *Figure 6A*) have access to oxygen, glucose, and ammonium, and perform either aerobic respiration or fermentation by overflow metabolism (*Basan et al., 2015*; *Cole et al., 2015*) – these two possible metabolic states cannot be distinguished with our current approaches. Cells at the bottom center of the colony (orange region in *Figure 6A*) are anaerobic yet they have access to glucose and ammonium from the agar-solidified medium. These cells ferment glucose and secrete alanine, primarily *via* AlaE. Although many amino acid exporters have been described for *E. coli* (*Pacheco et al., 2021*; *Prindle et al., 2015*; *Rani et al., 2007*), their functions have remained elusive under regular physiological conditions. Our data now reveal a function for the alanine exporter AlaE during biofilm growth. Secreted alanine diffuses through the colony and can only be utilized by oxic

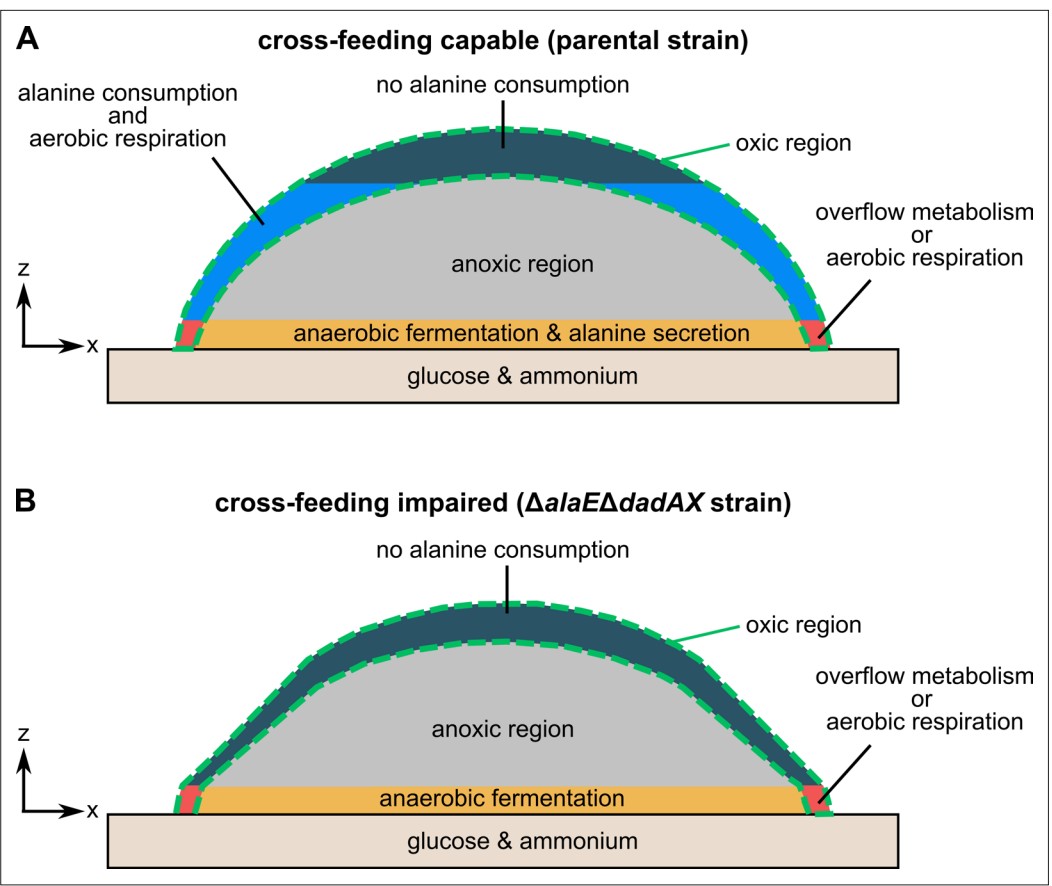

**Figure 6.** Model for alanine cross-feeding in *E. coli* colony biofilms. This model applies to colonies grown for 72 hr on solid M9 agar containing glucose and ammonium. (**A**) In cross-feeding capable colonies, cells in the bottom layer of the biofilm have access to glucose and ammonium. Only cells in the outer periphery of the biofilm (green dashed line) have access to oxygen. Cells in the red region can use ammonium, glucose, and oxygen to perform aerobic respiration or fermentation by overflow metabolism. Cells in the orange region have access to glucose and ammonium, but no oxygen. These cells secrete alanine. The secreted alanine can be consumed by cells in the oxic region above this layer (depicted as blue), which perform aerobic respiration and convert alanine into pyruvate and ammonium that can be used for growth and to maintain cell viability. (**B**) Colonies of *ΔalaEΔdadAX* cells have a reduced ability to consume and export alanine. These colonies have a region performing aerobic respiration or overflow metabolism (red), similar to the parental strain. These colonies also have an anoxic fermentation region (orange), yet this region displays significantly less alanine secretion compared to the parental strain. Furthermore, these colonies lack an alanine consuming population in the oxic region. Due to the limited alanine secretion and alanine consumption of this strain, *ΔalaEΔdadAX* colonies display higher cell death and less growth in the otherwise cross-feeding-dependent oxic region, resulting in a more conical colony shape in comparison to parental colonies.

nutrient-deprived cells (blue region in *Figure 6A*). Alanine consumption in the mid-height oxic region also has a detoxification effect, by reducing otherwise inhibitory levels of extracellular alanine. Alanine consumption at the oxic top of the colony is not significant, perhaps because the extracellular alanine is consumed before it reaches this region.

Colonies that are impaired in alanine cross-feeding because of a reduced ability to secrete and consume alanine display less growth and more cell death in the oxic glucose-deprived region, leading to a conical-shaped colony (*Figure 6B*). Furthermore, cross-feeding impaired cells are more susceptible to growth inhibition by high extracellular alanine levels (*Figure 4C*) and are outcompeted by the parental strain during colony growth.

The spatiotemporal organization of alanine metabolism during colony growth is unique among amino acids (*Figure 1D and E*), and is based on the secretion of alanine in the anaerobic, glucose-rich, and ammonium-rich base of the colony. Why is alanine secreted in this region? We speculate that

this is not an altruistic trait evolved to support a starving oxic subpopulation at a different location, because such a trait would be highly susceptible to social cheaters in a multi-species community. Instead, alanine secretion in this region may be a necessity to avoid high intracellular alanine levels that presumably result from anaerobic fermentation of glucose in the presence of ammonium. High alanine levels are inhibitory (*Figure 4C*), so that secretion of alanine and transport of alanine away from this population is beneficial to this population.

On the other side of the cross-feeding interaction, the alanine-consuming subpopulation in the aerobic mid-height region of the colony strongly benefits from the secreted alanine originating from the base of the colony. We note that the alanine consumption in the aerobic mid-height region of the colony necessarily causes a steeper alanine concentration gradient between the two interacting subpopulations, compared with a case in which no alanine is consumed. Due to Fick's law of diffusion, a steeper concentration gradient causes a higher diffusive flux. Therefore, the presence of the alanine-consuming subpopulation results in a benefit for the alanine-secreting subpopulation, by causing a higher diffusive flux of alanine away from the alanine-secreting population. It is unclear whether this benefit for the alanine-secreting subpopulation is significant, as it is not possible for us to measure local growth rates or cell viability in the anoxic base of the colony due to optical limitations of confocal fluorescence microscopy in this region. In summary, the interaction between the two cross-feeding subpopulations is likely mutualistic.

The spatial organization of alanine cross-feeding between two subpopulations we described in this study is analogous to the carbon cross-feeding in *E. coli* colonies *via* acetate, because it involves metabolite secretion in the anaerobic population and consumption in the carbon-starved oxic population (*Cole et al., 2015*; *Dal Co et al., 2019*; *Wolfsberg et al., 2018*). However, the location of the population that is proposed to consume acetate as a carbon source spans most of the oxic region of the colony (*Cole et al., 2015*), whereas alanine is primarily consumed only in the oxic mid-height region of the colony. Interestingly, our spatial transcriptomes did not reveal a signature for acetate cross-feeding between the anaerobic and oxic regions of the colony (*Figure 1—figure supplement 4B*), yet transcripts coding for enzymes involved in lactate, formate, and succinate metabolism display patterns that are indicative of spatially organized metabolism that could be the basis of carbon cross-feeding. Whether acetate, lactate, formate, and succinate are in fact cross-fed in our system remains to be tested in future work. In contrast to these metabolites, alanine can not only be used as carbon source, but also as a nitrogen source in the cross-feeding-dependent region, and we note that it is currently not clear what limits growth in the higher regions of the colony – whether it is carbon, nitrogen, or other elements such as iron, sulfur, or phosphorous. Very recently, it was shown that alanine can be shared in colonies of the Gram-positive bacterium *Bacillus subtilis*, and that this effect required three-dimensional colonies for unknown reasons (*Arjes et al., 2021*). In our *E. coli* model system, three-dimensional growth is required to create an anoxic region replete with carbon and nitrogen sources that causes alanine secretion, and we speculate that this effect may also be required in *B. subtilis*, and in other species.

Several cross-feeding interactions have been described in detail for multi-species communities (*Henson et al., 2019*; *Kim et al., 2017*; *Moree et al., 2012*; *Pande et al., 2016*; *Røder et al., 2020*; *Smith et al., 2019*; *Watrous et al., 2013*; *Yang et al., 2009*). Even though cross-feeding interactions are likely a ubiquitous process in single-species bacterial multicellular structures (*San Roman and Wagner, 2018*), only a few metabolic interactions between subpopulations have been documented for single-species biofilms (*Arjes et al., 2021*; *Cole et al., 2015*; *Evans et al., 2020*; *Lin et al., 2018*; *Liu et al., 2015*). For *E. coli* colonies grown on agar-solidified medium, and for bacterial communities in general, it is still unclear how many subpopulations interact metabolically, and on which length and time scales these interactions take place, and how significant these many interactions are for the community growth and stability.

## Conclusion

In this study, we employed an unbiased approach based on temporal and spatial transcriptomes and metabolomes to reveal that a multitude of amino acids and mixed acid fermentation pathways display profiles that are consistent with cross-feeding. Particularly strong regulation was displayed by alanine metabolism, and we showed that alanine is a cross-fed metabolite inside *E. coli* colonies, between two spatially segregated subpopulations, with an interaction length scale of tens of microns. Alanine

consumption supports growth in the cross-feeding-dependent region of the colony as a carbon and nitrogen source. Although many aspects of metabolism in biofilms are still unknown, methods for improved spatial and temporal analyses of metabolite profiles and transcriptome data promise the possibility to discover new metabolic interactions, and more generally understand the stability and functions of microbial communities.

# Materials and methods

**Key resources table**

| Reagent type (species) or resource | Designation | Source or reference | Identifiers | Additional information |
|---|---|---|---|---|
| Strain, strain background (*Escherichia coli*) | KDE261 | Drescher lab stock | Strain carrying plasmid pCP20 | Plasmid information listed in ***Supplementary file 2*** |
| Strain, strain background (*Escherichia coli*) | KDE262 | Drescher lab stock | Strain carrying plasmid pKD46 | Plasmid information listed in ***Supplementary file 2*** |
| Strain, strain background (*Escherichia coli*) | KDE264 | Drescher lab stock | Strain carrying plasmid pKD3 | Plasmid information listed in ***Supplementary file 2*** |
| Strain, strain background (*Escherichia coli*) | KDE265 | Drescher lab stock | Strain carrying plasmid pKD4 | Plasmid information listed in ***Supplementary file 2*** |
| Strain, strain background (*Escherichia coli*) | KDE1361 | Drescher lab stock | Strain carrying plasmid pNUT1361 | Plasmid information listed in ***Supplementary file 2*** |
| Strain, strain background (*Escherichia coli*) | KDE2338 | Drescher lab stock | Strain carrying plasmid pNUT2338 | Plasmid information listed in ***Supplementary file 2*** |
| Strain, strain background (*Escherichia coli*) | KDE2658 | Drescher lab stock, Addgene #64,969 | Strain carrying plasmid pUC18R6KT-mini-Tn7-Km | Plasmid information listed in ***Supplementary file 2*** |
| Strain, strain background (*Escherichia coli*) | KDE2659 | Drescher lab stock, Addgene #64,968 | Strain carrying plasmid pTNS2 | Plasmid information listed in ***Supplementary file 2*** |
| Strain, strain background (*Escherichia coli*) | KDE2674 | This study | Strain carrying plasmid pNUT2674 | Plasmid information listed in ***Supplementary file 2*** |
| Strain, strain background (*Escherichia coli*) | KDE2787 | This study | Strain carrying plasmid pNUT2787 | Plasmid information listed in ***Supplementary file 2*** |
| Strain, strain background (*Escherichia coli*) | KDE2838 | This study | Strain carrying plasmid pNUT2838 | Plasmid information listed in ***Supplementary file 2*** |
| Strain, strain background (*Escherichia coli*) | KDE474 | ***Serra et al., 2013a*** | *E. coli* AR3110 wild-type | |

*Continued on next page*

*Continued*

| Reagent type (species) or resource | Designation | Source or reference | Identifiers | Additional information |
|---|---|---|---|---|
| Strain, strain background (*Escherichia coli*) | KDE679 | *Vidakovic et al., 2018* | KDE474 (*E. coli* AR3110), P$_{tac}$-*mRuby2-mRuby2* and *Kan$^R$* inserted at *attB* site (P$_{tac}$ without operator). | |
| Strain, strain background (*Escherichia coli*) | KDE722 | *Vidakovic et al., 2018* | KDE679 with Δ*fliC*::scar. | |
| Strain, strain background (*Escherichia coli*) | KDE1899 | This study | KDE474 (*E. coli* AR3110) with Δ*fliC*::scar. | This strain can be obtained from the Drescher lab upon request |
| Strain, strain background (*Escherichia coli*) | KDE2007 | This study | KDE679 with Δ*fliC*::scar, Δ*alaE*::scar. | This strain can be obtained from the Drescher lab upon request |
| Strain, strain background (*Escherichia coli*) | KDE2009 | This study | KDE679 with Δ*fliC*::scar, Δ*dadAX*::scar. | This strain can be obtained from the Drescher lab upon request |
| Strain, strain background (*Escherichia coli*) | KDE2086 | This study | KDE679 with Δ*fliC*::scar, Δ*alaE*::scar, Δ*dadAX*::scar. | This strain can be obtained from the Drescher lab upon request |
| Strain, strain background (*Escherichia coli*) | KDE2183 | This study | KDE679 with Δ*fliC*::scar, Δ*cycA*::scar. | This strain can be obtained from the Drescher lab upon request |
| Strain, strain background (*Escherichia coli*) | KDE2185 | This study | KDE679 with Δ*fliC*::scar, Δ*livG*::scar. | This strain can be obtained from the Drescher lab upon request |
| Strain, strain background (*Escherichia coli*) | KDE2438 | This study | KDE679 with Δ*fliC*::scar, Δ*yaaJ*::scar. | This strain can be obtained from the Drescher lab upon request |
| Strain, strain background (*Escherichia coli*) | KDE2242 | This study | AR3110, P$_{tac}$-*sfgfp-sfgfp* and *Kan$^R$* inserted at *attB* site (P$_{tac}$ without operator), with Δ*fliC*::scar, Δ*alaE*::scar, Δ*dadAX*::scar. | This strain can be obtained from the Drescher lab upon request |
| Strain, strain background (*Escherichia coli*) | KDE2445 | This study | AR3110, P$_{tac}$-*sfgfp-sfgfp* and *Kan$^R$* inserted at *attB* site (P$_{tac}$ without operator), with Δ*fliC*::scar. | This strain can be obtained from the Drescher lab upon request |
| Strain, strain background (*Escherichia coli*) | KDE2533 | This study | KDE679 with Δ*fliC*::scar Δ*cycA*::scar, Δ*livG*::scar, Δ*alaE*::scar, Δ*yaaJ*::scar. | This strain can be obtained from the Drescher lab upon request |
| Strain, strain background (*Escherichia coli*) | KDE2564 | This study | KDE679 with Δ*fliC*::scar, Δ*cycA*::scar, Δ*dadAX*::scar. | This strain can be obtained from the Drescher lab upon request |
| Strain, strain background (*Escherichia coli*) | KDE2607 | This study | KDE679 with Δ*fliC*::scar, Δ*yaaJ*::scar, Δ*dadAX*::scar. | This strain can be obtained from the Drescher lab upon request |

*Continued on next page*

*Continued*

| Reagent type (species) or resource | Designation | Source or reference | Identifiers | Additional information |
|---|---|---|---|---|
| Strain, strain background (*Escherichia coli*) | KDE2937 | This study | KDE679 with Δ*fliC*::scar, P$_{tac}$-*sfgfp*(ASV) at the Tn7 insertion site, coding for an unstable superfolder GFP with an AANDENYAASV-tag. | This strain can be obtained from the Drescher lab upon request |
| Strain, strain background (*Escherichia coli*) | KDE2938 | This study | KDE679 with Δ*fliC*::scar, Δ*alaE*::scar, P$_{tac}$-*sfgfp*(ASV) at the Tn7 insertion site. | This strain can be obtained from the Drescher lab upon request |
| Strain, strain background (*Escherichia coli*) | KDE2939 | This study | KDE679 with Δ*fliC*::scar, Δ*dadAX*::scar, P$_{tac}$-*sfgfp*(ASV) at the Tn7 insertion site. | This strain can be obtained from the Drescher lab upon request |
| Strain, strain background (*Escherichia coli*) | KDE2940 | This study | KDE679 with Δ*fliC*::scar, Δ*alaE* Δ*dadAX*, P$_{tac}$-*sfgfp*(ASV) at the Tn7 insertion site. | This strain can be obtained from the Drescher lab upon request |
| Sequence-based reagent | KDO834 | This study | Insertions at the *attB* site | The sequence of this oligonucleotide can be found in ***Supplementary file 3*** |
| Sequence-based reagent | KDO894 | This study | *fliC* deletion | The sequence of this oligonucleotide can be found in ***Supplementary file 3*** |
| Sequence-based reagent | KDO895 | This study | *fliC* deletion | The sequence of this oligonucleotide can be found in ***Supplementary file 3*** |
| Sequence-based reagent | KDO1662 | This study | Insertions at the *attB* site | The sequence of this oligonucleotide can be found in ***Supplementary file 3*** |
| Sequence-based reagent | KDO2562 | This study | *alaE* deletion | The sequence of this oligonucleotide can be found in ***Supplementary file 3*** |
| Sequence-based reagent | KDO2563 | This study | *alaE* deletion | The sequence of this oligonucleotide can be found in ***Supplementary file 3*** |
| Sequence-based reagent | KDO2566 | This study | *dadAX* deletion | The sequence of this oligonucleotide can be found in ***Supplementary file 3*** |
| Sequence-based reagent | KDO2567 | This study | *dadAX* deletion | The sequence of this oligonucleotide can be found in ***Supplementary file 3*** |

*Continued on next page*

*Continued*

| Reagent type (species) or resource | Designation | Source or reference | Identifiers | Additional information |
|---|---|---|---|---|
| Sequence-based reagent | KDO2845 | This study | *cycA* deletion | The sequence of this oligonucleotide can be found in *Supplementary file 3* |
| Sequence-based reagent | KDO2846 | This study | *cycA* deletion | The sequence of this oligonucleotide can be found in *Supplementary file 3* |
| Sequence-based reagent | KDO2841 | This study | *livG* deletion | The sequence of this oligonucleotide can be found in *Supplementary file 3* |
| Sequence-based reagent | KDO2842 | This study | *livG* deletion | The sequence of this oligonucleotide can be found in *Supplementary file 3* |
| Sequence-based reagent | KDO3481 | This study | *yaaJ* deletion | The sequence of this oligonucleotide can be found in *Supplementary file 3* |
| Sequence-based reagent | KDO3482 | This study | *yaaJ* deletion | The sequence of this oligonucleotide can be found in *Supplementary file 3* |
| Sequence-based reagent | KDO3256 | This study | pNUT2338 construction | The sequence of this oligonucleotide can be found in *Supplementary file 3* |
| Sequence-based reagent | KDO3257 | This study | pNUT2338 construction | The sequence of this oligonucleotide can be found in *Supplementary file 3* |
| Sequence-based reagent | KDO3817 | This study | pNUT2838 construction | The sequence of this oligonucleotide can be found in *Supplementary file 3* |
| Sequence-based reagent | KDO3818 | This study | pNUT2838 construction | The sequence of this oligonucleotide can be found in *Supplementary file 3* |
| Sequence-based reagent | KDO3785 | This study | pNUT2674 construction | The sequence of this oligonucleotide can be found in *Supplementary file 3* |
| Sequence-based reagent | KDO3786 | This study | pNUT2674 construction | The sequence of this oligonucleotide can be found in *Supplementary file 3* |

*Continued on next page*

Continued

| Reagent type (species) or resource | Designation | Source or reference | Identifiers | Additional information |
|---|---|---|---|---|
| Sequence-based reagent | KDO4121 | This study | pNUT2787 construction | The sequence of this oligonucleotide can be found in *Supplementary file 3* |
| Sequence-based reagent | KDO4121 | This study | pNUT2787 construction | The sequence of this oligonucleotide can be found in *Supplementary file 3* |
| Sequence-based reagent | KDO4127 | This study | pNUT2838 construction | The sequence of this oligonucleotide can be found in *Supplementary file 3* |
| Software, algorithm | Matlab | MathWorks | Version R2019b | |
| Software, algorithm | Prism | GraphPad Software | Version 9.2.0 | |
| Software, algorithm | NIS-Elements | Nikon | Version 4.5.2 | |
| Software, algorithm | Inkscape | Inkscape | Version 1.0.1 | |
| Software, algorithm | CLC Genomics Workbench | Qiagen | Version 11.0 | |
| Chemical compound, drug | $Na_2HPO_4$ | Carl Roth | P030.2 | |
| Chemical compound, drug | $CoCl_2$ | Carl Roth | 7095.1 | |
| Chemical compound, drug | $MnSO_4$ | Sigma-Aldrich | M8179 | |
| Chemical compound, drug | $CuCl_2$ | Sigma-Aldrich | 307,483 | |
| Chemical compound, drug | $ZnSO_4$ | Sigma-Aldrich | Z0251 | |
| Chemical compound, drug | thiamine-HCl | Sigma-Aldrich | T4625 | |
| Chemical compound, drug | $FeCl_3$ | Sigma-Aldrich | 31,332 | |
| Chemical compound, drug | $MgSO_4$, | Sigma-Aldrich | M2643 | |
| Chemical compound, drug | $CaCl_2$ | Sigma-Aldrich | C5670 | |
| Chemical compound, drug | NaCl | Carl Roth | HN00.2 | |
| Chemical compound, drug | $(NH_4)_2SO_4$ | Carl Roth | 3746.2 | |
| Chemical compound, drug | $KH_2PO_4$ | Carl Roth | 3904.1 | |

## Strains, strain construction, and media

All *E. coli* strains used in this study are derivatives of the *E. coli* K-12 AR3110 strain (*Serra et al., 2013a*). For the construction of plasmids and bacterial strains, standard molecular biology techniques were applied (*Sambrook et al., 1989*), using enzymes purchased from New England Biolabs or Takara Bio. All AR3110 derivatives carried a constitutively expressed fluorescent protein expression system (based on the $P_{tac}$ promoter without the *lac* operator) inserted in the chromosome at the *attB* site. To generate chromosomal deletions, the lambda red system was used to replace the target region with an antibiotic cassette flanked by FRT sites (*Datsenko and Wanner, 2000*). Then, the Flp-FRT recombination system was utilized to remove the antibiotic resistance cassette (*Cherepanov and*

*Wackernagel, 1995*). To insert the unstable fluorescent protein sfGFP(ASV) (*Andersen et al., 1998*) expressed under the control of the P$_{tac}$ promoter (without the *lac* operator) and a chloramphenicol cassette flanked by FRT sites into the Tn7 insertion site, we used the protocol described by *Choi et al., 2005*. After the fragment was inserted, the chloramphenicol resistance cassette was removed using the Flp-FRT recombination system. All strains, plasmids, and oligonucleotides that were used in this study are listed in *Supplementary files 1-3*, respectively.

Cultures were grown in LB-Miller medium (10 g L$^{-1}$ NaCl, 10 g L$^{-1}$ tryptone, 5 g L$^{-1}$ yeast extract) for routine culture and cloning, or in M9 minimal salts medium supplemented with 5 g L$^{-1}$ D-glucose (which we refer to as "M9 medium" throughout the article for simplicity). The M9 minimal salts medium consisted of the following components: 42.3 mM Na$_2$HPO$_4$ (Carl Roth P030.2), 22 mM KH$_2$PO$_4$ (Carl Roth 3904.1), 11.3 mM (NH$_4$)$_2$SO$_4$ (Carl Roth 3746.2), 8.56 mM NaCl (Carl Roth HN00.2), 0.1 mM CaCl$_2$ (Sigma-Aldrich C5670), 1 mM MgSO$_4$, (Sigma-Aldrich M2643), 60 µM FeCl$_3$ (Sigma-Aldrich 31332), 2.8 µM thiamine-HCl (Sigma-Aldrich T4625), 6.3 µM ZnSO$_4$ (Sigma-Aldrich Z0251), 7 µM CuCl$_2$ (Sigma-Aldrich 307483), 7.1 µM MnSO$_4$ (Sigma-Aldrich M8179), 7.6 µM CoCl$_2$ (Carl Roth 7095.1). M9 agar plates were made using 8 mL of M9 medium (as defined above) with 1.5 % w/v agar-agar, aliquoted into petri dishes with 35 mm diameter and 10 mm height (Sarstedt 82.1135.500).

## Colony biofilm growth

Samples from -80 °C frozen stocks were used to inoculate LB-Miller medium with kanamycin (50 µg mL$^{-1}$) followed by incubation for 5 hr at 37 °C with shaking at 220 rpm. At this point, 1 µL of the culture was used to inoculate 5 mL of M9 medium inside a 100-mL-Erlenmeyer flask and grown at 37 °C with shaking at 220 rpm for 16–22 hr. The cultures were continuously kept in exponential phase by regular back-dilutions, with optical density at 600 nm (OD$_{600}$) always below 0.6. Aliquots from these cultures were passed through a sterile 0.45 µm pore size polyvinylidene fluoride membrane filter of diameter 5 mm, unless stated otherwise. High-resolution confocal fluorescence microscopy showed that this treatment resulted in spatially well-separated single cells on the filter membrane. Using clean and sterile stainless-steel tweezers, these filter membranes carrying the cells were immediately placed directly onto M9 agar plates and incubated at 37 °C for up to 72 hr.

## Microscopy

All imaging was performed using a Yokogawa spinning disk confocal unit mounted on a Nikon Ti-E inverted microscope. A Nikon 40 x air extra-long working distance objective with numerical aperture (NA) 0.60 was used for all imaging, except for mutant screening (*Figure 4—figure supplement 1*) and competition experiments (*Figure 4—figure supplement 4*) where a 4 x air objective (NA 0.13, Nikon) was used, and single-cell imaging (*Figure 2—figure supplement 1A*) were a 100 x oil objective (NA 1.45, Nikon) was used. All imaging was done inside a microscope incubator kept at 37 °C. Instead of imaging through the filter, colonies were imaged facing down and the Petri dish lid was removed (*Figure 1—figure supplement 1A*). To maintain a high humidity and avoid evaporation of the M9 agar, the space between the microscope objective and the petri dish was sealed with flexible plastic foil. To avoid condensation on the objective, the objective was heated using an objective heater at 37 °C.

## Colony detection and biovolume measurements with adaptive microscopy

In preliminary experiments, we observed that colony growth of the wild-type *E. coli* strain AR3110 resulted in heterogeneous shapes (*Figure 1—figure supplement 1B*) that were caused by flagella-based motility of cells on the filter membranes in the very early stages of incubation. To avoid the effects caused by cellular motility during early colony growth, all subsequently experiments were performed using a strain that lacks the flagellin FliC (Δ*fliC*), which we used as parental strain in this study. This strain was incapable of swimming motility, and the resulting colonies were highly reproducible in shape and highly symmetric (*Figure 1B* and *Figure 1—figure supplement 1C*).

To determine the biomass of colonies, the colonies were grown on filter membranes on M9 agar as explained above, but with the addition of 0.2 µm dark red fluorescent beads (Invitrogen, F8807) to the bacterial suspension prior to filtering. This resulted in far-red fluorescent beads being located on the filter in addition to the red fluorescent cells. Using adaptive microscopy (*Jeckel and Drescher, 2021*)

these beads were used to find the correct focal plane for imaging of the bacterial cells (or base of the colonies). After 12 , 18 , 24 , 32 , 42 , 48 , 60 , and 72 hr of colony growth, the colonies were imaged in 3D using confocal microscopy. All colonies on the membrane filter were imaged using an adaptive microscopy approach (*Jeckel and Drescher, 2021*) as follows: The whole filter was scanned using 2D imaging, followed by an identification of all colonies on the filter, followed by a high-resolution 3D imaging of each colony. From the scans of the whole filter, the number of colonies was determined. From the 3D images of the colonies, the biovolume was calculated. To obtain the biovolume, the outline of the colony was identified by thresholding the image gradient in each *z*-slice. The convex area of this binary image was then used as a measure for the biomass present in this plane such that summation over all slices followed by multiplication with the appropriate μm³/voxel calibration yielded the biovolume of each colony. The code used for image analysis is available on Github, https://github.com/knutdrescher/colonymetabolism.

## Quantification of mRuby2 fluorescence in colonies from images

The colonies were grown for 72 hr and then imaged using the microscopy conditions described above, and colonies were detected in the images as described above. The colony surface was defined as the interface between the colony and the air. Measurements of fluorescence levels inside the colony at different locations was performed analogous to methods that are available within the BiofilmQ software (*Hartmann et al., 2021*), yet some modifications were performed as described below, which made it more convenient to use separate functions (available on Github, https://github.com/knutdrescher/colonymetabolism).

To measure the vertical and horizontal fold changes in mRuby2 fluorescence as a function of the distance to the colony surface (*Figure 2A*) two different approaches were applied. To measure the fluorescence in a vertical direction inside the colony, a vertical cylinder of radius 3.3 μm was defined in the center of the colony. The fluorescence in that cylinder was quantified and the values obtained were normalized to the fluorescence at the colony surface. To measure the fold change in fluorescence horizontally within the colony, a *xy*-horizontal plane at the base of the colony was selected. The mRuby2 fluorescence was quantified in the plane and normalized to the fluorescence at the colony-air interface within the same plane.

The fold-change in mRuby2 fluorescence as a function of the distance to the colony surface in whole colonies was also measured irrespective of the orientation of the measurement axis (*Figure 2—figure supplement 1D*). For this, the mRuby2 fluorescence intensity values of all locations inside the colony with a similar distance to the colony surface (see definition above) were averaged.

To determine the total fraction of fluorescent fraction of cells in a colony (insert of *Figure 2B*), colonies were grown as described above for either 12 h, 18h, 24 h, 32 h, 42 h, 48h, 60 h, or 72 h. Then, the mRuby2 fluorescence in the colonies was imaged using confocal microscopy, using the microscopy conditions described above. Inside the colony, 3D regions that were fluorescent in the mRuby2 channels were identified *via* thresholding. The fluorescent fraction was calculated as the fluorescent volume divided by the total volume of the colony.

## Liquid growth assays

Samples from -80 °C frozen stocks were used to inoculate LB-Miller medium, followed by incubation for 5 hr at 37 °C with shaking at 220 rpm. Each culture was back-diluted 5,000-fold into 5 mL of M9 medium inside a 100-mL-Erlenmeyer flask, and grown at 37 °C with shaking at 220 rpm. At an $OD_{600}$ of 0.3 each culture was washed three times in M9 minimal salts medium (lacking glucose and ammonium sulfate) and resuspended in the same volume of the medium of interest. These bacterial suspensions were diluted 10-fold and transferred into a 96-well plate (Sarstedt, 82.1581.001), and incubated at 37 °C with shaking in a microtiter plate reader (Epoch2, Biotek). The resulting growth curve data was analyzed using Matlab (version R2019b, Mathworks).

To simultaneously measure $OD_{600}$ and the ratio between unstable GFP (with the ASV-tag) and mRuby2 as a function of time (*Figure 5—figure supplement 1*), a glycerol stock of the strain KDE2937 was used to inoculate LB-Miller medium, followed by incubation for 5 hr at 37 °C with shaking at 220 rpm. The culture was then back-diluted 5000-fold into 5 mL of M9 medium inside a 100 mL Erlenmeyer flask and grown for 16 hr. This culture was used to inoculate 75 mL of M9 medium inside a 1 L Erlenmeyer flask with an adjusted $OD_{600}$ of 0.05. This culture was grown at 37 °C with shaking at

220 rpm. Aliquots of the culture were taken every 30 min, to measure the $OD_{600}$ and to determine the ratio between unstable GFP (with the ASV-tag) and mRuby2 using microscopy. To image the aliquots, they were placed between a cover slip and a M9 agar pad and imaged as described in the microscopy methods section.

## Anoxic liquid growth assays

Samples from -80 °C frozen stocks were used to inoculate LB-Miller medium, followed by incubation for 5 hr at 37 °C with shaking at 220 rpm. Each culture was then back-diluted 5000-fold into 5 mL of M9 medium inside a 100 mL Erlenmeyer flask, and grown at 37 °C with shaking at 220 rpm. At $OD_{600}$ = 0.7 each culture was washed three times in M9 minimal salts medium (lacking glucose and ammonium sulfate) and resuspended in 300 µL. These bacterial suspensions were used to inoculate 20 mL of the particular medium under investigation with a starting $OD_{600}$ of 0.03, which was placed into a 50 mL bottle that was closed with a gas-tight lid containing a rubber septum (Duran, 292062803). Immediately after inoculation, the bottle and the culture were made anoxic using the following protocol: A needle connected to a system that can switch between applying a vacuum and gaseous $N_2$ was introduced through the rubber septum. Then, vacuum was applied to remove the gas in the bottle for 1 min, followed by flushing the bottle with $N_2$ at 1 bar for 1 min, while the medium in the bottle was continuously mixed using a magnetic stirrer. This process was repeated for 15 cycles. After this process, the needle was slowly removed from the bottle and the cultures were incubated at 37 °C with shaking at 220 rpm. After 96 hr of incubation, the $OD_{600}$ was measured (*Figure 3D*).

## Liquid culture conditions for measurements of extracellular alanine in supernatants

Samples from -80 °C frozen stocks were used to inoculate LB-Miller medium with kanamycin (50 µg mL⁻¹), followed by incubation for 5 hr at 37 °C with shaking at 220 rpm. The cultures were then back-diluted 5000-fold into 5 mL of M9 medium inside a 100-mL-Erlenmeyer flask and incubated at 37 °C with shaking at 220 rpm. These cultures were kept in exponential phase by regular back-dilutions, with optical density at 600 nm ($OD_{600}$) always below 0.6. At an $OD_{600}$ of 0.3, the cultures were centrifuged at 14000 g for 2 min and the supernatant was removed. To investigate aerobic growth conditions, the pellets were suspended in M9 minimal salts medium supplemented with glucose and the $OD_{600}$ was adjusted to 0.1 by dilution. For oxic starvation conditions, the pellets were suspended in same volume of M9 medium lacking ammonium sulfate but including glucose, M9 medium lacking glucose but including ammonium sulfate, or M9 medium lacking both glucose and ammonium sulfate. The cultures were then placed into 100-mL-Erlenmeyer flasks at 37 °C with shaking at 220 rpm as before. To investigate anoxic conditions, the pellets were suspended in the same media as described above, but the resulting cultures were transferred into a closed 15-mL-conical centrifuge tube (Sarstedt, 62.554.100) filled to the top. The tubes were oriented horizontally and incubated at 37 °C with shaking at 100 rpm. For the oxic and anoxic starvation conditions, the samples were taken after 2 hr of incubation. For aerobic and anaerobic conditions that permitted growth, samples were taken during exponential growth phase. Samples were processed and analysed using mass spectrometry as described below.

## Sample processing for mass spectrometry-based metabolomics

To measure metabolites in whole colonies over time, filter membranes that carried colonies were transferred into 150 µL of a mixture of 40:40:20 (v/v) acetonitrile:methanol:water at –20 °C for metabolite extraction. This suspension was vortexed with a glass bead to disrupt the colonies.

To measure extracellular metabolites from colonies, the filter membranes carrying the colonies were resuspended in 1 mL phosphate-buffered saline (PBS; 8 g l⁻¹ NaCl, 0.2 g ¹⁻¹ KCl, 1.44 g l⁻¹ $Na_2HPO_4$, 0.24 g l⁻¹ $KH_2PO_4$, pH 7.4) at 37 °C. In this case, no glass bead vortexing was needed, as the colonies readily dissolved. The suspension was immediately vacuum-filtered using a 0.45 µm pore size filter (HVLP02500, Merck Millipore) and 100 µL of the flow-through were mixed with 400 µL of a mixture of 50:50 (v/v) acetonitrile:methanol at –20 °C.

To measure extracellular metabolites from liquid cultures, 1 mL of grown cultures were filtered on a 0.45 µm pore size filter (HVLP02500, Merck Millipore) and 100 µL of the flow through were mixed with 400 µL of a mixture of 50:50 (v/v) acetonitrile:methanol at –20 °C.

All extracts were centrifuged for 15 min at 11000 g at –9 °C and stored at –80 °C until mass spectrometry analysis.

## Mass spectrometry measurements

For metabolomics, centrifuged extracts were mixed with $^{13}$C-labeled internal standards. Chromatographic separation was performed on an Agilent 1290 Infinity II LC System (Agilent Technologies) equipped with an Acquity UPLC BEH Amide column (2.1 × 30 mm, particle size 1.8 µm, Waters) for acidic conditions and an iHilic-Fusion (P) HPLC column (2.1 × 50 mm, particle size 5 µm, Hilicon) for basic conditions. The following binary gradients with a flow rate of 400 µl min$^{-1}$ were applied. Acidic condition: 0–1.3 min isocratic 10% A (water with 0.1% v/v formic acid, 10 mM ammonium formate), 90 % B (acetonitrile with 0.1% v/v formic acid), 1.3–1.5 min linear from 90% to 40% B; 1.5–1.7 min linear from 40% to 90% B, 1.7–2 min isocratic 90 % B. Basic condition: 0–1.3 min isocratic 10% A (water with formic acid 0.2 % (v/v), 10 mM ammonium carbonate), 90 % B (acetonitrile); 1.3–1.5 min linear from 90% to 40% B; 1.5–1.7 min linear from 40% to 90% B, 1.7–2 min isocratic 90 % B. The injection volume was 3.0 µl (full loop injection).

Ions were detected using an Agilent 6495 triple quadrupole mass spectrometer (Agilent Technologies) equipped with an Agilent Jet Stream electrospray ion source in positive and negative ion mode. The source gas temperature was set to 200 °C, with 14 L min$^{-1}$ drying gas and a nebulizer pressure of 24 psi. Sheath gas temperature was set to 300 °C and the flow to 11 L min$^{-1}$. Electrospray nozzle and capillary voltages were set to 500 and 2500 V, respectively. Metabolites were identified by multiple reaction monitoring (MRM). MRM parameters were optimized and validated with authentic standards (*Guder et al., 2017*). Metabolites were measured in $^{12}$C and $^{13}$C isoforms, and the data was analysed with a published Matlab code (*Guder et al., 2017*).

## Data analysis for metabolomics

For whole-colony metabolite measurements, the mass spectrometry measurements were normalized by the total biovolume measured with confocal microscopy. For measurements of the extracellular metabolites in colonies (*Figure 4B*), the mass spectrometry measurements were normalized by colony number and average colony volume determined by confocal microscopy. To create heatmaps of the metabolite dynamics, the Genesis software (*Sturn et al., 2002*) was used.

## Whole-colony transcriptomes

Colonies were grown on filter membranes as described above. After 12 , 18 , 24 , 32 , 42 , 48 , 60 , and 72 hr of growth,the filters carrying the colonies were picked up using plastic forceps and transferred into 1.5 mL Eppendorf tubes, which were immediately placed into liquid nitrogen, followed by storage at –80 °C until further processing. To extract the RNA, a glass bead and 600 µL of cell lysis buffer were added during thawing of the sample at room temperature. Cell lysis buffer consisted of TE (10 mM Tris, adjusted to pH 8.0 with HCl, 1 mM EDTA) and 1 mg/mL of chicken egg lysozyme (Sigma, L6876). The colonies were disrupted by vortexing and the cell suspension was moved to a new 1.5 mL Eppendorf tube. Then, total RNA was extracted using the hot SDS/phenol method (*Jahn et al., 2008*) with some modifications as follows. Cells were lysed at 65 °C for 2 min in the presence of 1 % (w/v) SDS, and the lysate was incubated with 750 µL of Roti-Aqua-Phenol (Carl Roth, A980) at 65 °C for 8 min, followed by the addition of 750 µL chloroform (Sigma, C2432) to the aqueous phase and centrifugation using a phase lock gel tube (VWR, 733–2478). RNA was purified from this suspension by ethanol precipitation and dissolved in 60 µL of RNase-free water. Samples were then treated with TURBO DNase (Thermo Fisher, AM2238) and rRNA depletion was performed using Ribo-Zero rRNA Removal Kit for bacteria (Illumina, MRZB12424). Sequencing library preparation was carried out using NEBNext Ultra II Directional RNA Library Prep with Sample Purification Beads (NEB, E7765S). Sequencing was carried out at the Max Planck Genome Centre (Cologne, Germany) using an Illumina HiSeq3000 with 150 bp single reads. All transcriptomic analyses were performed using the software CLC Genomics Workbench v11.0 (Qiagen). The *E. coli* K-12 W3310 genome (*Hayashi et al., 2006*), parental strain of *E. coli* AR3110 (*Serra et al., 2013a*), was used as reference for annotation. For creating heatmaps, clustering of transcripts was performed using the software Genesis (*Sturn et al., 2002*).

## Spatial transcriptomes

Filter membranes carrying colonies that were grown for 72 hr were transferred into 2 mL Eppendorf tubes (one filter per tube). Cells from the colonies on a filter were then quickly suspended in 1 mL PBS by vortexing and pipetting, followed by removal of the filter from the tube, and fixation of the cells by adding formaldehyde (Sigma, F8775) to a final concentration of 4%, for 10 min at room temperature. Formaldehyde fixation did not affect the transcriptomic profile or the fluorescence intensity (*Figure 2—figure supplement 2*). Then, the fixed cells were washed three times with PBS and the cell suspension was filtered with a 5 μm pore size filter (Sartorius, 17594) to remove aggregates. Cells were separated using fluorescence-activated cell sorting (BD FACSAria Fusion), using the mRuby2 fluorescence as a signal. After sorting, approximately $10^6$ cells were collected in 10 mL of PBS for each bin. To concentrate the samples, they were vacuum-filtered using a 0.45 μm membrane filter (Millipore, HVLP02500). The filters containing the cells were suspended in 400 μL of the cell lysis buffer (same composition as above). The suspension was frozen in liquid nitrogen and stored at –80 °C until RNA extraction.

Total RNA was extracted using the same protocol as described above (based on the hot SDS/phenol method), but with the following modifications to minimize the number of steps and loss of RNA: after treatment with phenol, the aqueous and organic phases were both directly transferred to a phase lock gel tube (VWR, 733–2478) without centrifugation. Chloroform was added and centrifuged at 15000 rpm for 15 min at 12 °C. After centrifugation, RNA in the aqueous phase was purified and collected in 10 μL of RNase-free water using Agencourt RNAClean XP (Beckman Coulter, A63987) according to the manufacturer's recommendations. Samples were then treated with TURBO DNase (Thermo Fisher, AM2238) and rRNA depletion was performed using Ribo-Zero rRNA Removal Kit for bacteria (Illumina, MRZB12424). Library preparation was carried out using NEBNext Ultra II Directional RNA Library Prep with Sample Purification Beads (NEB, E7765S). Sequencing was carried out at the Max Planck Genome Centre (Cologne, Germany) using an Illumina HiSeq3000 with 150 bp single reads.

Spatial and temporal transcriptomic data were uploaded to the Gene Expression Omnibus (GEO) repository with the accession code GSE175768.

## Measurements of the fraction of dead cells

To measure the fraction of dead cells within the oxic region of the colonies, the colonies were grown on filter membranes on M9 agar as described above, but with the exception that SYTOX Green (Thermofisher, S7020) with a final concentration of 2.5 μM was added to the M9 agar plates. SYTOX Green is a nucleic acid stain that can only diffuse into dead cells with a compromised cell membrane. After incubation for 72 hr, the mRuby2 fluorescence and the SYTOX Green fluorescence in the colonies was imaged using confocal microscopy. Inside the colony, 3D regions that were fluorescent in both the mRuby2 and SYTOX Green channels were identified via thresholding. The fraction of dead cells was calculated as the thresholded volume present in both the mRuby2 and SYTOX Green channels, divided by the thresholded volume of the mRuby2 channel. For the quantification of the fraction of dead cells in the oxic region of the colony, only cells located within 30 μm from the outer surface of the colony were considered, because only in this region oxygen penetration was high enough to generate a sufficiently strong mRuby2 signal.

To measure the fraction of dead cells in liquid cultures, cultures were grown in M9 medium, which was supplemented with SYTOX Green (2.5 μM), incubated at 37 °C inside 96-well plates with continuous shaking. The $OD_{600}$ and SYTOX Green fluorescence were measured in a microtiter plater reader (Spark 10 M, Tecan). The fraction of dead cells was calculated as the SYTOX Green fluorescence normalized by the $OD_{600}$.

## Strain competition experiments

Samples from -80 °C frozen stocks were used to inoculate LB-Miller medium with kanamycin (50 μg mL$^{-1}$), followed by incubation for 5 hr at 37 °C with shaking at 220 rpm. The cultures were then back-diluted 5000-fold into 5 mL of M9 medium inside a 100-mL-Erlenmeyer flask and incubated at 37 °C with shaking at 220 rpm until exponential phase. Then, the cultures were adjusted to an $OD_{600}$ of 0.05 and two strains were mixed in a 1:1 ratio and the solution was diluted 100-fold. For each experiment, one strain expressed the fluorescent protein mRuby2 and the other strain the fluorescent protein

sfGFP. The exact ratio of the strains in the inoculation suspension was measured using flow cytometry. Then, 1 µL of this mixture was filtered on a membrane filter, which it was placed on solid M9 agar and incubated at 37 °C. After 72 hr, the colonies were imaged and the ratio of each strain was measured in a ring with a width of 35 µm and at a distance of 350 µm from the edge of the inoculation spot using basic image analysis and image thresholding.

## Oxygen measurements

Oxygen concentrations were measured in 72-hr-old colonies (grown on M9 agar as described above) using a 25-µm-tip oxygen microsensor (Unisense OX-25) according to the manufacturer's instructions. Briefly, the oxygen microsensor was calibrated using a two-point calibration. The microsensor was first calibrated to atmospheric oxygen using a calibration chamber (Unisense CAL300) containing water continuously bubbled with air. The microsensor was then calibrated to a 'zero' oxygen point using an anoxic solution of 0.1 M sodium ascorbate and 0.1 M sodium hydroxide in water. Oxygen measurements were then taken through the top ≥100 µm of the colony biofilm in 5-µm-steps using a measurement time of 3 s at each position, and a wait time between measurements of 5 s. A micromanipulator (Unisense MM33) was used to move the microsensor within the colony. Profiles were recorded using a multimeter (Unisense) and the SensorTrace Profiling software (Unisense).

## Image analysis scripts

All Matlab scripts used for image analysis are available on Github, https://github.com/knutdrescher/colonymetabolism (copy archived at swh:1:rev:cfc8d212071e634b6b7aa8462b745a14761193e9, *Drescher, 2021*).

## Statistical analysis

Statistical analysis was carried out using GraphPad Prism v8 (GraphPad Software), except for the statistical analysis for transcriptomic data, which was performed using the software CLC Genomics Workbench 11.0 (Qiagen). All statistical tests performed and sample sizes are reported in the corresponding figure legend. Each biological replicate was performed on a different day.

## Acknowledgements

We are grateful to Thorben Schramm and Nicole Paczia for performing additional metabolite measurements. We thank Gabriele Malengo for helping with the cell sorting procedure. We also thank Miriam Bayer, Praveen K Singh, Daniel Rode, Lucia Vidakovic, Sanika Vaidya, and all members of the Drescher lab for their comments and fruitful discussions. This work was supported by grants from the European Research Council (StG-716734), Deutsche Forschungsgemeinschaft (SFB987 and DR 982/5–1), Minna James Heineman Foundation, Bundesministerium für Bildung und Forschung (TARGET-Biofilms), Max Planck Society, and the Swiss National Science Foundation NCCR "AntiResist" (to K.D.), and an NSF CAREER award and NIH/NIAID grant R01AI103369 (to L.E.P.D.).

## Additional information

### Funding

| Funder | Grant reference number | Author |
| --- | --- | --- |
| H2020 European Research Council | StG-716734 | Knut Drescher |
| Deutsche Forschungsgemeinschaft | SFB987 | Hannes Link Knut Drescher |
| Deutsche Forschungsgemeinschaft | DR 982/5-1 | Knut Drescher |
| Heineman Foundation | | Knut Drescher |

| Funder | Grant reference number | Author |
|---|---|---|
| Bundesministerium für Bildung und Forschung | TARGET-Biofilms | Knut Drescher |
| Max-Planck-Gesellschaft | | Knut Drescher |
| Schweizerischer Nationalfonds zur Förderung der Wissenschaftlichen Forschung | NCCR AntiResist | Knut Drescher |
| National Science Foundation | CAREER Award | Lars EP Dietrich |
| National Institute of Allergy and Infectious Diseases | R01AI103369 | Lars EP Dietrich |

The funders had no role in study design, data collection and interpretation, or the decision to submit the work for publication.

## Author contributions

Francisco Díaz-Pascual, Conceptualization, Data curation, Formal analysis, Investigation, Methodology, Writing – original draft, Writing – review and editing, Project administration, Visualization; Martin Lempp, Data curation, Investigation, Methodology, Validation; Kazuki Nosho, Investigation, Methodology, Validation; Hannah Jeckel, Data curation, Formal analysis, Investigation, Software; Jeanyoung K Jo, Formal analysis, Investigation, Methodology; Konstantin Neuhaus, Raimo Hartmann, Methodology, Software; Eric Jelli, Software; Mads Frederik Hansen, Investigation, Methodology; Alexa Price-Whelan, Writing – review and editing; Lars EP Dietrich, Hannes Link, Funding acquisition, Resources, Supervision, Writing – review and editing; Knut Drescher, Conceptualization, Funding acquisition, Methodology, Project administration, Resources, Supervision, Visualization, Writing – original draft, Writing – review and editing

## Author ORCIDs

Francisco Díaz-Pascual (iD) http://orcid.org/0000-0001-8947-682X
Kazuki Nosho (iD) http://orcid.org/0000-0002-4811-1397
Hannah Jeckel (iD) http://orcid.org/0000-0002-7080-4907
Jeanyoung K Jo (iD) http://orcid.org/0000-0003-1543-1148
Konstantin Neuhaus (iD) http://orcid.org/0000-0002-9877-5765
Raimo Hartmann (iD) http://orcid.org/0000-0002-4924-6402
Eric Jelli (iD) http://orcid.org/0000-0001-6202-2449
Mads Frederik Hansen (iD) http://orcid.org/0000-0001-9283-304X
Alexa Price-Whelan (iD) http://orcid.org/0000-0001-7587-7534
Lars EP Dietrich (iD) http://orcid.org/0000-0003-2049-1137
Knut Drescher (iD) http://orcid.org/0000-0002-7340-2444

## Decision letter and Author response

Decision letter https://doi.org/10.7554/eLife.70794.sa1
Author response https://doi.org/10.7554/eLife.70794.sa2

# Additional files

## Supplementary files

• Supplementary file 1. Bacterial strains used in this study. Abbreviations: Kan = kanamycin. Superscript "R" = resistance. "-" = fusion. "::" = insertion. The scar corresponds to 5'-GAAGTTCC TATACTTTCTAGAGAATAGGAACTTC-3' sequence.

• Supplementary file 2. Plasmids used in this study. Abbreviations: Kan = kanamycin, Amp = ampicillin, Chl = chloramphenicol. Superscript "R" = resistance. "-" = fusion.

• Supplementary file 3. DNA oligonucleotides used in this study.

• Transparent reporting form

## Data availability

Spatial and temporal transcriptomic data were uploaded to the Gene Expression Omnibus (GEO) repository with the accession code GSE175768. All data generated or analysed during this study are included in the manuscript and supporting files. All Matlab scripts used for image analysis are available on Github, https://github.com/knutdrescher/colonymetabolism (copy archived at https://archive.softwareheritage.org/swh:1:rev:cfc8d212071e634b6b7aa8462b745a14761193e9).

The following dataset was generated:

| Author(s) | Year | Dataset title | Dataset URL | Database and Identifier |
|---|---|---|---|---|
| Diaz-Pascual F, Nosho K, Drescher K | 2021 | Spatial alanine metabolism determines local growth dynamics of *Escherichia coli* colonies | https://www.ncbi.nlm.nih.gov/geo/query/acc.cgi?acc=GSE175768 | NCBI Gene Expression Omnibus, GSE175768 |

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
