## [Editor Report]

In this manuscript the authors characterize the temporal and spatial distribution of
cells with different metabolic states within colony of biofilms of the model
bacterium *Escherichia coli* using a combination of transcriptomics,
metabolomics, and quantitative measurements of growth. They show that within the
biofilm cells performing different metabolic functions are distributed in different
regions of the colonies, and propose a model where nutrient cross-feeding through
the amino acid alanine explains the phenotypic heterogeneity observed within the
biofilm. Interestingly, they show that mutants impaired in cross-feeding have a
fitness disadvantage. The finding reported and the innovative technical approaches
used, can potentially be applied to identify novel metabolic interactions between
communities of bacteria and understand how bacterial subpopulations interact
spatially and metabolically within biofilms.

---

## [Decision Letter]

**Decision letter after peer review:**

Thank you for submitting your article "Spatial alanine metabolism determines
local growth dynamics of *Escherichia coli* colonies" for
consideration by *eLife*. Your article has been reviewed by 3 peer
reviewers, including Karina B Xavier as the Reviewing Editor and Reviewer #1, and
the evaluation has been overseen by Gisela Storz as the Senior Editor.

Essential revisions:

In general, all three reviewers found the work technical very sound with potential
impact for thinking about the spatial structure of bacterial species in biofilm in
all bacterial species. While they recognize the interest of the proposed nutrient
cross-feeding model for explaining the observed phenotypic spatial heterogeneity
within the biofilm the reviewers think that additional experiments or arguments are
needed to support the proposed model. Additionally, the reviewers also propose a
couple of additional simple experiments which could help improve the significance of
the finding reported in the paper.

The major points to be addressed are listed in the 3 essential revisions below.
Additionally, you can also find in the end of this letter the sperate
"recommendations for the authors" written by each of the reviewers which
should help you to improve the manuscript and to better understand the rationale
used by each of the reviewers that support for the proposed essential reviews. We
recommend that you take the recommendations written by each of the reviewers into
consideration while preparing the revised version, but you do not need to reply to
each of the individual comments from the reviewers, you will only need to reply the
essential points summarized here by the reviewing editor:

1) We propose that the authors revise the model considering the following point:
Specifically, we are concern about the fact that the authors are assuming that the
cells are respiring, or fermenting based on whether they are in aerobic (oxygen
expose) or anaerobic areas of the biofilm or not. In liquid cultures it is well
known that bacteria use fermentative metabolism to promote fast growth, irrespective
of oxygen levels, if the glucose levels in the medium are high enough. This could
happen at the edge of the biofilm; cells are in close contact to the glucose of the
agar medium. Thus, we question that the fast-growing bacteria at the edge of the
biofilm are probably fermenting cells and not respiring cells.

Therefore, we recommend that the authors revisit their model or include data that
support the correlation between oxygen/respiration. One possibility that we
recommend to the authors is to correlate the fermenting or respiring biofilm regions
to glucose (and or alanine) levels instead of oxygen levels (this point is discussed
in more detail in the specific recommendation of reviewer #3).

2) Also related to the cross-feeding model, we recommend that the model described in
Figure 6 also addresses what happens with the alaE dadAX mutant (which is impaired
in cross-feeding), as this could help in understanding the overall proposed model
(see major comment 1 from reviewer 1).

3) To improve the broader relevance and overall implication of the findings described
in the paper the reviewers propose the following two set of experiments: A) to
determine if there are potential fitness advantages of the proposed alanine
cross-feeding mechanism the authors could ask if there is a fitness advantage for
the WT versus the mutant impaired in cross-feeding within the biofilm (see major
comment 2 from reviewer 1), B) to shed some light on potential selective pressures
related to alanine metabolism the authors could look at biofilms in colonies
composed of a mixtures of WT and the alaE dadAX mutant (which is impaired in
cross-feeding) and look for potential spatial sorting of the two strains within the
biofilm community (see major comment 4 of reviewer 2).

*Reviewer #1 (Recommendations for the authors):*

I have a couple major comments/questions that could be important for strengthening
the model proposed in the paper:

1 – The authors present a cross-feeding model in Figure 6 based on alanine
cross-feeding explaining with the proposed model for the WT biofilm colonies. But, I
think that the discussion about what happens in the alaE dadAX mutant is also
important. I think it is important to also discuss what happens with the mutant,
namely regarding cell death within the biofilm. Perhaps also have a scheme for the
mutant, but at least to address it in the discussion.

2 – I think it is not clearly demonstrated that there is a benefit for the overall
fitness of the population within colony biofilm for having the proposed
cross-feeding mechanism. But because there is more cell death in the alaE dadAX
mutant it is likely that there is a fitness benefit. This could be tested by
scrapping WT and mutant biofilms, disaggregating them and then determine colony
forming units. This experiment should allow to address this question.

3 – Alanine secretion under high nitrogen and high carbon conditions resemble what
happens with acetate secretion, namely also because both metabolites are then
consumed by starved cells. The authors mention this in the discussion. I think that
the fact that cells are secreting alanine and not acetate is interesting. Perhaps it
is relevant to comment on the fact that the cells secreting alanine in the biofilm
have an excess of both Carbon and Nitrogen, as opposed to carbon only. Perhaps cells
secrete acetate when Carbon is in excess but Nitrogen and secrete alanine when both
are in excess, no?

*Reviewer #2 (Recommendations for the authors):*

Most of my suggestions focus on the broader relevance of these findings, which I
think the authors could extend quite substantially with a few more experiments.

1. Line 100: the study suggests that the shift in the transcriptome after 24 h is due
to oxygen depletion, but the authors don't actually show this (I don't think). They
could monitor oxygen directly over time to address this point directly, e.g. through
oxygen-sensitive nanoparticles.

2. Line 154: I'm confused: couldn't the mRuby2 fluorescence increase due to moving
the colony, which would expose it to more oxygen? Also, there's a typo: "in for
formerly".

3. In terms of why the mutant cells are dying (line 207), the authors suggest that it
is due to the accumulation of alanine, which they can recapitulate through live/dead
staining in liquid culture. It could be fascinating to track cells at the
single-cell level – do the cells lyse? (if so, this could impact surrounding cells
in a colony, by releasing more alanine and thereby increasing the local
concentration) Or do the cells simply halt growth and become permeable?

4. In a colony that is made up of a mixture of wild-type and ∆alaE∆dadAX cells, do
the two strains spatially sort based on their differential ability to grow in
aerobic regions of the colony? This simple experiment could shed some light on the
selective pressures on alanine metabolism.

5. Line 229: can it simply be cell death that is providing the exogenous alanine?

6. Figure 4D: why is there not increased cell death in the parent at high
concentrations (5 mM) since the growth rate is lower at this concentration? And more
generally, is the reduction in growth rate in the mutant quantitatively explained by
the death rate, or do you have to invoke a second mechanism?

7. Line 275: For the bulge morphology, can this be restored in a mixed colony of
wild-type and the mutant? Or do you get an intermediate phenotype?

8. How is the spatial patterning that they see affected by the ability of cells to
produce matrix?

9. Their model seems to predict what you would see in a more two-dimensional text,
when colonies are sandwiched between a cover glass. This could be a neat means to
further investigate the significance of alanine export for growth rate of the
colony.

10. What about the presence of another species that consumes alanine, will it
colocalize preferentially at the aerobic region?

*Reviewer #3 (Recommendations for the authors):*

I recommend the authors to revisit the idea of linking oxic/anoxic regions with
alanine producers/consumers. Maybe is true but it needs stronger evidence-support
because is different from what is described in liquid cultures. In the light of the
data presented, I think is better idea to link alanine metabolism to glucose
availability; it makes a stronger case,

1) The authors equate the aerobic or anaerobic areas of the biofilm with respiring
(fast-growing) or fermenting (slow-growing) cells, respectively. Is this already
known? I am confused because, in liquid cultures, cells use fermentation to grow
when carbohydrates are abundant and rely on respiration when the concentration of
carbohydrates decreases. Hence, the regions with fast cell growth should show
fermentative growth (alanine production) whereas the regions with slow growth
undergo respiration and consume alanine as a non-preferred catabolite. Notice that
these two biofilm areas are equally exposed to oxygen. This does not really fit with
the model in figure 6 in which fast growing cells respire at high carbohydrate
concentration. I can see a correlation between alanine metabolism and glucose
abundance but the connection to oxygen levels is not clear to me.

2) I am not convinced that maturation of the fluorescent protein mRuby2 is indicative
of respiring bacteria. I think it is a very elegant way to identify the aerobic
areas of the biofilm but again, bacteria can ferment even in the presence of oxygen,
if the concentration of carbohydrates is high enough (i.e. crabtree effect in yeast,
overflow metabolism in bacteria or Warburg effect in cancer cells). The
subpopulation of fluorescence cells likely combine fermenting (closer edge to the
agar) and respiring (middle-height edge of the colony) bacteria. Moreover, the
authors removed glucose from the medium to alter the metabolic processes that
consume oxygen (line 151) but this will severely impact fermentative growth as
well.

3) I do not understand how these extracellular alanine concentrations are toxic to
the cells. The paper cited is about intracellular alanine (Katsube et al., 2019). Is
there anything known about how extracellular alanine levels are toxic to the
cells?

Line 114: "Lactate, formate and succinate biosynthesis, however, displayed
interesting dynamics during colony growth". This sentence has no meaning as it
does not describe the data. In what way are these dynamics interesting?

Line 125: "All amino acid abundances decreased during colony growth, except for
alanine, which remained relatively constant with a peak abundance at 32 h".
Several aminoacids derive from alanine metabolism. I wonder how the level of these
aminoacids did not remain closer to that of alanine

Line 129: I do not understand how the transcriptomic or metabolomic data invited the
authors to explore whether alanine metabolism is spatially heterogeneous during
biofilm growth.

Lines 217-260: these two paragraphs describing the phenotypes of different mutants in
alanine metabolism are hard to digest. I read them several times and I do not think
I understood.

---

## [Author Response]

Essential revisions:1) We propose that the authors revise the model considering the following point:
Specifically, we are concern about the fact that the authors are assuming that
the cells are respiring, or fermenting based on whether they are in aerobic
(oxygen expose) or anaerobic areas of the biofilm or not. In liquid cultures it
is well known that bacteria use fermentative metabolism to promote fast growth,
irrespective of oxygen levels, if the glucose levels in the medium are high
enough. This could happen at the edge of the biofilm; cells are in close contact
to the glucose of the agar medium. Thus, we question that the fast-growing
bacteria at the edge of the biofilm are probably fermenting cells and not
respiring cells.Therefore, we recommend that the authors revisit their model or include data that
support the correlation between oxygen/respiration. One possibility that we
recommend to the authors is to correlate the fermenting or respiring biofilm
regions to glucose (and or alanine) levels instead of oxygen levels (this point
is discussed in more detail in the specific recommendation of reviewer #3).

We thank the reviewers for this comment. We agree that our statements equating oxic
conditions to respiration were inaccurate and that the cells in the glucose-,
ammonium-, and oxygen-rich region of the colony (i.e. the edge of the base of the
colony) could perform overflow metabolism instead of aerobic respiration. We
currently cannot distinguish these two metabolic states in the relevant region of
the colony, yet such a distinction would not have a major impact on our conclusions.
To correct our inaccurate statements, we have therefore made the following changes
to the manuscript:

– In Figure 6, we labeled the relevant region of the colony with “overflow metabolism
or aerobic respiration”, and we also updated the caption of Figure 6
accordingly.

– We also added a new statement in the main text where the model (i.e. Figure 6) is
described, as follows (main text, lines 344-348):

“Based on our results, we propose the following model for the spatial organization of
alanine metabolism in colonies that have grown for 72 h (Figure 6A): Cells at the
bottom periphery of the colony (red region in Figure 6A) have access to oxygen,
glucose and ammonium, and perform either aerobic respiration or overflow metabolism
(Basan et al., 2015; Cole et al., 2015) – these two possible metabolic states cannot
be distinguished with our current approaches.”

– Throughout the entire manuscript, we also revised all mentions of the terms
“aerobic”, “anaerobic”, “respiration”, and “fermentation”, and, where appropriate,
rephrased them.

2) Also related to the cross-feeding model, we recommend that the model described
in Figure 6 also addresses what happens with the alaE dadAX mutant (which is
impaired in cross-feeding), as this could help in understanding the overall
proposed model (see major comment 1 from reviewer 1).

We agree that this would be a helpful addition to the manuscript. In response, we
have made the following changes to the manuscript:

– We expanded the model description in the Discussion section, to include the
Δ*alaE*Δ*dadAX* mutant (in lines 358-362).

– We have included a new panel in Figure 6 to highlight the differences between the
parental strain and the Δ*alaE*Δ*dadAX* mutant.

3) To improve the broader relevance and overall implication of the findings
described in the paper the reviewers propose the following two set of
experiments: A) to determine if there are potential fitness advantages of the
proposed alanine cross-feeding mechanism the authors could ask if there is a
fitness advantage for the WT versus the mutant impaired in cross-feeding within
the biofilm (see major comment 2 from reviewer 1), B) to shed some light on
potential selective pressures related to alanine metabolism the authors could
look at biofilms in colonies composed of a mixtures of WT and the alaE dadAX
mutant (which is impaired in cross-feeding) and look for potential spatial
sorting of the two strains within the biofilm community (see major comment 4 of
reviewer 2).

We thank the reviewers once more for their helpful suggestions.

Regarding proposed experiment A:

The reviewers are asking for an experiment to test if there is a fitness benefit for
the whole colony if the colony is capable of the spatially organized alanine
cross-feeding process we are describing in this manuscript. To investigate this
question, we have measured the colony diameter and colony height of the
cross-feeding capable strain (parental strain) and the cross-feeding impaired strain
(Δ*alaE*Δ*dadAX*). The results show that the
colony size is not significantly different between these two strains (panels B,C in
Figure 4—figure supplement 1). We expect that CFU counts of such colonies (as
suggested by reviewer 1, comment 2) would have a larger measurement uncertainty than
microscopy-based measurements of colony sizes so that we performed microscopy
measurements instead of CFU measurements. To be able to resolve the expected small
fitness benefit of cross-feeding for the whole colony, a very large number of
biological replicates would be needed, given the natural variation of colony sizes
between biological replicates shown in Figure 4—figure supplement 1.

We also note that in our system the cell death of the
Δ*alaE*Δ*dadAX* colonies reaches only 4% of the
total biomass in the cross-feeding dependent region of the colony (shown in Figure
4A). We therefore expect that for the whole colony, the global fitness benefit of
alanine cross-feeding is relatively small. To highlight this point in the
manuscript, we have made the following changes:

– We revised the section of the manuscript where the effects of alanine secretion and
consumption on colony growth are described (lines 224-230):

“To determine how interference with alanine export and consumption affects colony
growth, we created individual and combinatorial deletions of known alanine transport
and degradation genes. None of these deletions affected the cellular growth rates in
liquid culture (Figure 4—figure supplement 1A). These deletions also did not cause
clear phenotypes in colony height or diameter after 72 h of incubation on M9 agar
(Figure 4—figure supplement 1B,C), indicating that alanine export and consumption do
not have large effects on global colony size. However, the mutants displayed
substantial differences when we measured the fraction of dead cells in the oxic
region”

– We also expanded the section of the manuscript on the effect of alanine
cross-feeding on colony morphology (lines 315-321):

“From measurements of the colony height and diameter for mutants impaired in
cross-feeding, we know that alanine cross-feeding does not have a major influence on
colony size (Figure 4—figure supplement 1). These measurements of colony height and
diameter also show that alanine cross-feeding does not contribute to aerobic growth
at the very top of the colony or at the outer edge of the base. We therefore
investigated effects of alanine cross-feeding on cellular growth in the oxic region
at mid-height,”

– Figure 4—figure supplement 1 and caption have been updated.

Regarding proposed experiment B:

We agree that this experiment would provide helpful new insights into the impact of
alanine cross-feeding on colony biofilm growth. We have performed this experiment
and added a new supplemental figure and a new section in the manuscript to describe
the results.

It is important to note that the proposed experiment required that two strains were
mixed together and a drop of the resulting culture suspension was spotted onto a
filter membrane on agar. This inoculation condition is different from all other
colonies grown and investigated for this study – these colonies were exclusively
grown from a single bacterial cell seeded onto the filter membrane on agar. The
resulting 3D morphologies of colonies grown from a single cell and colonies grown
from a 1 µL drop of culture suspension are quite different. In particular, the
colonies grown from a drop of culture suspension were flatter and wider compared to
the colonies grown from single cells. Therefore, the results from the two-strain
competition experiments are not completely comparable to the other experiments in
this study, yet they provide some new insights that we believe are valuable.

In our implementation of the two-strain competition experiments, we mixed two strains
together at approximately 1:1 ratio, measured the exact strain ratio using flow
cytometry, and inoculated a small drop of this suspension on a filter membrane on M9
agar. We then measured the strain frequency at the growing front of the colony using
microscopy.

Our results show that the cross-feeding impaired
Δ*alaE*Δ*dadAX* strain was outcompeted by the
parental strain in colonies (new Figure 4—figure supplement 4), despite having no
liquid culture growth difference (Figure 4—figure supplement 1).

In response to this comment by the reviewers, we have made the following changes to
the manuscript:

– We added a new paragraph in the Results section of the manuscript (lines
296-312).

– We also briefly mention these new results in the discussion (lines 358-362).

– We also added a new figure: Figure 4—figure supplement 4.

Reviewer #1 (Recommendations for the authors):I have a couple major comments/questions that could be important for
strengthening the model proposed in the paper:1 – The authors present a cross-feeding model in Figure 6 based on alanine
cross-feeding explaining with the proposed model for the WT biofilm colonies.
But, I think that the discussion about what happens in the alaE dadAX mutant is
also important. I think it is important to also discuss what happens with the
mutant, namely regarding cell death within the biofilm. Perhaps also have a
scheme for the mutant, but at least to address it in the discussion.

This comment is reflected in the Essential Revision #2, and we provide a detailed
answer to this comment above.

2 – I think it is not clearly demonstrated that there is a benefit for the
overall fitness of the population within colony biofilm for having the proposed
cross-feeding mechanism. But because there is more cell death in the alaE dadAX
mutant it is likely that there is a fitness benefit. This could be tested by
scrapping WT and mutant biofilms, disaggregating them and then determine colony
forming units. This experiment should allow to address this question.

This comment is reflected in the Essential Revision #3 (“Experiment A”), and we
provide a detailed answer to this comment above.

3 – Alanine secretion under high nitrogen and high carbon conditions resemble
what happens with acetate secretion, namely also because both metabolites are
then consumed by starved cells. The authors mention this in the discussion. I
think that the fact that cells are secreting alanine and not acetate is
interesting. Perhaps it is relevant to comment on the fact that the cells
secreting alanine in the biofilm have an excess of both Carbon and Nitrogen, as
opposed to carbon only. Perhaps cells secrete acetate when Carbon is in excess
but Nitrogen and secrete alanine when both are in excess, no?

This is an interesting suggestion. We agree that there are some analogies between
alanine secretion and acetate secretion. Although our spatial transcriptome data
show no characteristic signature of acetate cross-feeding, it is not impossible that
acetate cross-feeding takes place in our conditions (perhaps no spatially organized
transcriptional regulation is required for this). Cole et al., 2015 (https://doi.org/10.1186/s12918-015-0155-1) present data that is
consistent with acetate cross-feeding in *E. coli* colonies, using
fluorescent reporters. In our system fluorescent protein reporters for acetate
secretion at the base of the colony would not be conclusive because of the oxygen
requirement for fluorescent protein folding – which was the basis of our method for
obtaining spatial transcriptomes.

To the best of our knowledge acetate secretion does not rely on low nitrogen levels,
and we speculate that cells in the anoxic region are perhaps secreting acetate as
well as alanine (and perhaps also succinate, formate, and lactate, and likely even
more compounds). We are currently planning a follow-up study, where we will
investigate more broadly whether these (and other) metabolites are cross-fed.

In response to this comment, we have added a statement in the discussion (lines
390-395):

“Interestingly, our spatial transcriptomes did not reveal a signature for acetate
cross-feeding between the anaerobic and oxic regions of the colony (Figure 1— figure
supplement 4B), yet transcripts coding for enzymes involved in lactate, formate, and
succinate metabolism display patterns that are indicative of spatially organized
metabolism that could be the basis of carbon cross-feeding. Whether acetate,
lactate, formate and succinate are in fact cross-fed in our system remains to be
tested in future work.”

Reviewer #2 (Recommendations for the authors):Most of my suggestions focus on the broader relevance of these findings, which I
think the authors could extend quite substantially with a few more
experiments.1. Line 100: the study suggests that the shift in the transcriptome after 24 h is
due to oxygen depletion, but the authors don't actually show this (I don't
think). They could monitor oxygen directly over time to address this point
directly, e.g. through oxygen-sensitive nanoparticles.

We have performed such measurements, which are summarized below, including the
context:

The temporal transcriptome data we acquired (Figure 1C) is an average of the entire
colony and does not resolve heterogeneity inside the colony. The transcriptome shift
that occurs at around 24 h shows that the average transcriptome of the colony has a
shift towards anaerobic metabolism, which is a state in which the average
transcriptome remains until the end of our experiments (72 h).

For 72 h colonies we performed direct measurements of oxygen levels with spatial
resolution using an oxygen microsensor. These measurements of oxygen levels inside
the colony closely align with mRuby2 fluorescence levels (mRuby2 requires oxygen to
fold into a fluorescent conformation), indicating that mRuby2 levels in our system
correlate with oxygen abundance (Figure 2A), and that mRuby2 fluorescence can be
used to measure spatial heterogeneity of oxygen levels in our system.

By measuring mRuby2 levels in space and time during colony growth, and by computing
the ratio between mRuby2 fluorescent/non-fluorescent cells we can compute the ratio
of cells in the oxic/anoxic regions of the colony (Figure 2B, inset). Indeed, these
measurements of heterogeneity of oxygen levels show a shift towards a largely anoxic
colony around 24 h.

To highlight these measurements, we expanded the description of Figure 2B, resulting
in the following changes to the manuscript (lines 148-152):

“During colony growth, the fraction of fluorescent cells in the colony decreased
(inset in Figure 2B), and the majority of the colony became non-fluorescent (i.e.
anoxic) around 24 h, which coincides with the time at which the whole-colony
transcriptome shifted towards anaerobic metabolism (Figure 1C). This decrease in the
mRuby2-fluorescent population during colony growth ultimately lead to a thin layer
of fluorescent cells in the air-facing part of the colony (Figure 2B).”

The reviewer’s suggestion of an alternative measurement method for oxygen levels is
interesting, but we feel that the existing data we described above are already
unambiguous regarding the change in metabolism related to a spatially segregated
anaerobic population.

2. Line 154: I'm confused: couldn't the mRuby2 fluorescence increase due to
moving the colony, which would expose it to more oxygen? Also, there's a typo:
"in for formerly".

We thank the reviewer for pointing out that our description was not clear. Indeed,
there are two effects when the filter carrying the colonies is moved to a fresh M9
agar plate without glucose:

– Effect 1: The fresh agar plate initially has some oxygen underneath the
colonies.

– Effect 2: The cells in the colony are now starved of glucose and therefore consume
a lot less oxygen, which enables oxygen to diffuse into the colony to enable mRuby2
folding throughout the entire colony.

Effect 2 is shown in Figure 2—figure supplement 1B,C,D. Effect 1 will only cause an
initial folding of mRuby2 near the bottom surface of the colony (similar to the thin
mRuby2 fluorescent zone in the air-exposed region of the colony), but this effect
cannot explain that mRuby2 fluorescence emerges throughout the entire depth of the
colony after transfer to the M9 agar plate without glucose (which is shown in the
newly added Figure 2 – supplement 1C).

To clarify these two effects in the manuscript, we have revised the relevant section
of the control experiment description (lines 163-172):

“To test this, we starved the colonies by transferring the filter membrane carrying
the colonies to an M9 agar plate lacking glucose, which strongly decreases the
colony capacity to consume oxygen. We observed that in this case, mRuby2 proteins
that are located in the formerly dark anoxic region of the colony became fluorescent
(Figure 2—figure supplement 1B,C,D). The oxygen in the fresh agar plate is not able
to cause the entire colony to become fluorescent without the reduced oxygen
consumption in the colony caused by the lack of glucose. The finding that the entire
colony becomes fluorescent after the transfer is consistent with the interpretation
that in the absence of glucose, cells consume less oxygen so that molecular oxygen
can penetrate into the colony to enable chromophore maturation of mRuby2 in the
formerly anoxic region (Figure 2—figure supplement 1B,C,D).”

We have also corrected the typo mentioned by the reviewer.

3. In terms of why the mutant cells are dying (line 207), the authors suggest
that it is due to the accumulation of alanine, which they can recapitulate
through live/dead staining in liquid culture. It could be fascinating to track
cells at the single-cell level – do the cells lyse? (if so, this could impact
surrounding cells in a colony, by releasing more alanine and thereby increasing
the local concentration) Or do the cells simply halt growth and become
permeable?

This is an interesting idea. Optical imaging inside the colonies on agar is
unfortunately at lower resolution than our imaging of biofilms on glass surfaces in
microfluidic chambers, where single-cell resolution is achievable. In microfluidic
chambers, we can use oil-immersion objectives which have a very high resolution. The
lower imaging resolution in colonies on agar is due to the requirement to either use
air immersion objectives or water-immersion objectives that don’t have the required
resolution for single-cell imaging in dense *E. coli* aggregates (not
even near the edges).

Therefore, we cannot see whether the cells definitely lyse, although their membrane
is compromised, as inferred from the live/dead stain. However, we know from liquid
culture experiments (Figure 4D) in which extracellular alanine was added that the
optical density did not substantially decrease (an OD decrease would be expected
following lysis), even though the live/dead stain showed an increase. This suggests
that the majority of dead cells in the colony don’t lyse.

Moreover, when the concentration of extracellular alanine was determined for the
different mutant colonies (Figure 4B), we observed that the Δ*dadAX*
strain presented an increased level of extracellular alanine. However, the strain
does not display an increased cell death phenotype in colonies (Figure 4A) so that
it is unlikely that the increased levels of extracellular alanine originate from
lysed or dead cells.

To highlight these points in the manuscript, we made the following changes:

– Lines 248-250:

“The high extracellular alanine levels of the ΔdadAX colonies are unlikely to be
caused by permeable or lysed cells, as the ΔdadAX colonies do not display elevated
levels of cell death (Figure 4A).”

– Lines 279-283:

“we found that in the presence of high extracellular alanine concentrations,
ΔalaEΔdadAX mutants displayed higher cell death levels than the parental strain,
which was particularly strong in stationary phase conditions (Figure 4D). The
increased cell death of the ΔalaEΔdadAX mutants was not accompanied by a reduction
in optical density, indicating these cells did not lyse.”

4. In a colony that is made up of a mixture of wild-type and ∆alaE∆dadAX cells,
do the two strains spatially sort based on their differential ability to grow in
aerobic regions of the colony? This simple experiment could shed some light on
the selective pressures on alanine metabolism.

This comment is related to Essential Revision #3 (“Experiment B”), and we provide a
detailed answer to this comment above.

5. Line 229: can it simply be cell death that is providing the exogenous
alanine?

Once again, we thank the reviewer for this important comment that we addressed above
(in the answer to Major questions/suggestions #3). Additionally, we added a sentence
in the manuscript in lines 248-250 to clarify this.

6. Figure 4D: why is there not increased cell death in the parent at high
concentrations (5 mM) since the growth rate is lower at this concentration? And
more generally, is the reduction in growth rate in the mutant quantitatively
explained by the death rate, or do you have to invoke a second mechanism?

It is indeed thought provoking that the parental strain displays a reduction in
growth rate but not an increased cell death at 5 mM extracellular alanine (Figure
4C, D) – even though 5 mM extracellular alanine does lead to both a reduction in
growth rate and increased cell death for the *ΔalaEΔdadAX* mutant.
The results from the parental strain show that apparently the reduction in growth
rate is not caused by cell death. We speculate that a reduced growth rate is
measurable before the stress is large enough to cause cell death and that a second
mechanism for these two phenomena does not need to be invoked. We note that the
parental strain still has the capability to decrease their intracellular alanine
levels via the proteins encoded by *dadAX* and *alaE.*
Our data in Figure 4C, D suggests that these proteins help the parental strain to
survive the increased alanine levels, and under some conditions this strain even
uses extracellular alanine to grow (Figure 4D).

To clarify our results and interpretation on this topic, we have now included
additional explanations in lines 274-281:

“To test this hypothesis, we measured cell viability for the parental strain and
mutants in liquid cultures with and without exogenous alanine during midexponential
phase and in stationary phase. We found that even though the parental strain
displayed a reduced growth rate when exposed to extracellular alanine (Figure 4C),
the parental strain did not display increased levels of cell death in such
conditions (Figure 4D) – likely due to their ability to secrete and consume alanine,
allowing them to control their intracellular levels of alanine. In contrast, we
found that in the presence of high extracellular alanine concentrations, ΔalaEΔdadAX
mutants displayed higher cell death levels than the parental strain”

7. Line 275: For the bulge morphology, can this be restored in a mixed colony of
wild-type and the mutant? Or do you get an intermediate phenotype?

This would be nice, but unfortunately, is not possible to mix 2 (or more) strains and
obtain a colony morphology that is directly comparable to colonies that were grown
from a single cell. In all of our experiments (except for the mixed-strain
experiments that were performed in response to your comment #4, our new Figure
4—figure supplement 4) we grew colonies from a single bacterial cell. To obtained
mixed-strain colonies, it is necessary to inoculate the colonies with a small drop
of culture suspension (containing the different strains), which necessarily results
in a different initial inoculation condition. The resulting colonies in mixed-strain
experiments are wider and flatter compared with colonies grown from a single
bacterium. Therefore, the experiment proposed by the reviewer would unfortunately
not be interpretable.

To clarify the difference between colonies grown from a mixture of strains and
colonies grown from a single bacterium, we included an explanatory sentence in lines
298-303:

“To test if this is the case, we generated pairwise mixtures of different strains and
inoculated these mixtures onto a membrane filter placed on M9 agar. It is important
to note that this inoculation procedure using a liquid drop (leading to colonies
that were inoculated by hundreds of cells) creates different colony morphologies
compared to colonies grown from a single bacterium (which was the growth condition
for all other experiments with colonies in this article).”

8. How is the spatial patterning that they see affected by the ability of cells
to produce matrix?

Although matrix genes are expressed (mentioned in lines 110-111), we speculate that
during the first 72 h of colony growth (which is the time window we study in this
article), and for colonies grown from a single bacterial cells at 37°C, the matrix
does not yet have a major influence on the colony morphology. The pyramid-like
colony morphology we observed for the *ΔalaEΔdadAX* mutant resembles
the simulations from Warren et al., 2019 (Terence Hwa´s lab), in which colony growth
was simulated without the presence of an extracellular matrix. We also note that the
primary matrix component responsible for wrinkling *of E. coli*
colonies (curli amyloid fibers) are not strongly expressed at 37°C (mentioned
initially by Olsén et al., 1989, and corroborated by several groups).

Studies by Regine Hengge et al., have revealed that for the same strain we use, but
cultured at lower temperatures (which induces stronger production of curli amyloid
fibers), and inoculated from a drop of culture suspension, the matrix does have a
very large effect on colony morphology after longer incubation times.

9. Their model seems to predict what you would see in a more two-dimensional
text, when colonies are sandwiched between a cover glass. This could be a neat
means to further investigate the significance of alanine export for growth rate
of the colony.

In fact, we think that our experiments show that the three-dimensionality of the
colonies is important for the alanine cross-feeding, because we need the
oxygen-gradient and the opposing glucose and ammonium gradients (only the anoxic
glucose- and ammonium-rich cells secrete alanine). We now explicitly mention this in
the Discussion section (in lines 400-403).

To replicate such a spatially organized alanine metabolism in a two-dimensional
microfluidic system, we predict that it would require a microfluidic chip that
produces opposing gradients in oxygen and glucose and ammonium.

10. What about the presence of another species that consumes alanine, will it
colocalize preferentially at the aerobic region?

This is an interesting idea and we agree with the reviewer’s prediction – for species
that can only consume alanine in aerobic conditions (which is true for *E.
coli*, Figure 3D). Such a mixed-species experiment would require
detailed characterization of the other species’ preference for alanine over glucose
in oxic and anoxic conditions, to interpret results from mixed-species colony growth
experiments. Although we agree that this would be an interesting direction to
dissect interactions in a simple multi-species community, our opinion is that such
an experiment would go beyond the scope of the current study. We also note that the
first author of this study has departed from the lab now, which means that such an
experiment would not be possible within a reasonable time frame.

Reviewer #3 (Recommendations for the authors):I recommend the authors to revisit the idea of linking oxic/anoxic regions with
alanine producers/consumers. Maybe is true but it needs stronger
evidence-support because is different from what is described in liquid cultures.
In the light of the data presented, I think is better idea to link alanine
metabolism to glucose availability; it makes a stronger case,1) The authors equate the aerobic or anaerobic areas of the biofilm with
respiring (fast-growing) or fermenting (slow-growing) cells, respectively. Is
this already known? I am confused because, in liquid cultures, cells use
fermentation to grow when carbohydrates are abundant and rely on respiration
when the concentration of carbohydrates decreases. Hence, the regions with fast
cell growth should show fermentative growth (alanine production) whereas the
regions with slow growth undergo respiration and consume alanine as a
non-preferred catabolite. Notice that these two biofilm areas are equally
exposed to oxygen. This does not really fit with the model in figure 6 in which
fast growing cells respire at high carbohydrate concentration. I can see a
correlation between alanine metabolism and glucose abundance but the connection
to oxygen levels is not clear to me.

This is a valuable comment. In our initial submission the description of the
metabolic state of the oxic region of the base of the colony (which has high glucose
levels) was inaccurate – it is certainly possible that the cells in this region
ferment through overflow metabolism. We have therefore corrected our description of
this region in Figure 6 and in the manuscript main text. This comment is related to
Essential Revision #1 above, where we provide a detailed answer including the
specific changes we made to the manuscript in response to this comment.

Regarding the reviewer’s comment about alanine secretion and glucose availability: In
liquid cultures we only detected alanine secretion in anoxic conditions (Figure 3A).
The oxic, fast-growing liquid culture conditions did not result in measurable
extracellular alanine levels in our conditions. These results are consistent with
the interpretation that only anoxic fermenting cells secrete alanine. However, from
our current data we cannot conclude whether or not cells in our fast-growing liquid
culture conditions performed overflow metabolism, because our metabolite
measurements did not include acetate for technical reasons. In response to this
comment, we carefully revised the paragraph in which Figure 3A is described (lines
205213):

“The spatial transcriptomes suggest that alanine is primarily secreted in the anoxic
region of the biofilm. To test this, we explored under which combination of
carbon/nitrogen/oxygen availability *E. coli* secretes alanine in
shaking liquid conditions. Mass spectrometry measurements from culture supernatants
clearly showed that alanine is only secreted under anoxic conditions with glucose
and ammonium (Figure 3A), which is an environment that corresponds to the anoxic
base of the colony, where cells are in contact with the glucose- and ammonium-rich
M9 agar. Oxic conditions with abundant glucose and ammonium did not result in
significant alanine secretion. This finding suggests that alanine is secreted in the
anoxic base of the colony, which is consistent with the spatial transcriptome
results.”

2) I am not convinced that maturation of the fluorescent protein mRuby2 is
indicative of respiring bacteria. I think it is a very elegant way to identify
the aerobic areas of the biofilm but again, bacteria can ferment even in the
presence of oxygen, if the concentration of carbohydrates is high enough (i.e.
crabtree effect in yeast, overflow metabolism in bacteria or Warburg effect in
cancer cells). The subpopulation of fluorescence cells likely combine fermenting
(closer edge to the agar) and respiring (middle-height edge of the colony)
bacteria. Moreover, the authors removed glucose from the medium to alter the
metabolic processes that consume oxygen (line 151) but this will severely impact
fermentative growth as well.

We are grateful that the reviewer pointed out this inaccuracy in our terminology. We
agree that fluorescence of mRuby2 is not indicative of respiration. We used mRuby2
fluorescence levels as an indication of oxic regions. To correct this inaccuracy in
our terminology, we have carefully gone through every mentioning of the words
“respiration”, “fermentation”, “aerobic”, and “anaerobic” and, where appropriate,
rephrased the relevant sentences based on the terms “oxic” and “anoxic” throughout
the manuscript.

3) I do not understand how these extracellular alanine concentrations are toxic
to the cells. The paper cited is about intracellular alanine (Katsube et al.,
2019). Is there anything known about how extracellular alanine levels are toxic
to the cells?

The experiments in which we varied the extracellular alanine level and observed
growth inhibition (Figure 4C) show that the extracellular alanine concentration has
a large effect. In particular, cells that were unable to export alanine
(Δ*alaE*) were more affected by extracellular alanine compared to
the parental strain, but the double mutant
(Δ*dadAX*Δ*alaE*) which has an impaired ability to
secrete or degrade alanine, was most affected. These results support the idea that
the extracellular alanine levels can affect the intracellular alanine levels.

As far as we are aware, the molecular mechanism for how alanine can be inhibitory to
the cells is currently not described in the literature. Perhaps alanine toxicity is
linked to misloading of tRNAs or the modification of the direction of some enzymatic
reactions due to an excess of alanine.

In response to this comment, we added the following clarifications in the manuscript
(lines 258-262):

“For cells that lack both the major alanine exporter AlaE and the major alanine
degradation pathway via DadA and DadX, we hypothesized that the presence of
extracellular alanine might lead to an accumulation of intracellular alanine to
toxic levels. It has previously been shown that excess levels of intracellular
alanine can inhibit growth (Katsube, Ando, and Yoneyama, 2019), yet the molecular
mechanism underlying this process is still unclear.*”*

Line 114: "Lactate, formate and succinate biosynthesis, however, displayed
interesting dynamics during colony growth". This sentence has no meaning as
it does not describe the data. In what way are these dynamics interesting?

We agree that this sentence lacked meaning. We have now modified the sentence to
(lines 120-122):

“However, transcripts of the lactate, formate, and succinate biosynthesis pathways
displayed differential regulation during colony growth (Figure 1—figure supplement
4A).”

Line 125: "All amino acid abundances decreased during colony growth, except
for alanine, which remained relatively constant with a peak abundance at 32
h". Several aminoacids derive from alanine metabolism. I wonder how the
level of these aminoacids did not remain closer to that of alanine

We thank the reviewer for this interesting comment. We are not completely sure why is
this the case. We offer two speculations:

– Speculation 1: Figure 1E reports measurements of the total amino acid pool in the
colony (the sum of intracellular and extracellular levels of each amino acid) – it
is possible that only alanine is secreted and therefore the measured alanine levels
primarily reflect the extracellular levels, whereas the other amino acids remain
intracellular and they decrease during colony growth because cells adapt to an
increasingly nutrient-poor environment.

– Speculation 2: Several amino acids can be made from pyruvate. These reactions
generally require multiple enzymes, except for alanine, which can be converted to
and from pyruvate in a simple enzymatic reaction. Furthermore, several amino acids
can be converted into alanine. The close link of alanine levels to central
metabolism (and in particular to pyruvate levels) combined with its secretion during
anaerobic fermentation could be the origin of the difference between the alanine
profile and the profile of other amino acids.

Because these arguments are quite speculative, we did not include them in the
manuscript.

Line 129: I do not understand how the transcriptomic or metabolomic data invited
the authors to explore whether alanine metabolism is spatially heterogeneous
during biofilm growth.

We thank the reviewer for highlighting that there was a logical gap at this point in
the manuscript. We have now added an additional sentence in lines 136-139 to bridge
the logical gap in the argument, as follows:

“Biofilms are expected to be metabolically heterogeneous so that we hypothesized that
alanine metabolism could be spatially organized inside biofilms. To test this
hypothesis, we developed a method to measure transcriptomes with spatial resolution
in the colonies.”

Lines 217-260: these two paragraphs describing the phenotypes of different
mutants in alanine metabolism are hard to digest. I read them several times and
I do not think I understood.

We carefully read these paragraphs again, and discussed them with colleagues who are
not co-authors of the manuscript and detected a logical break in the mentioning of
the -alanine exporter, which we have now corrected by rearranging the sentences and
addition additional text. We have also added a summary sentence at the end of the
first paragraph, to improve the readability. We also made numerous smaller edits in
these paragraphs aimed at improving the readability.